# Splat and Distill: Augmenting Teachers with Feed-Forward 3D Reconstruction For 3D-Aware Distillation

**David Shavin & Sagie Benaim**
Department of Computer Science
The Hebrew University of Jerusalem
Jerusalem, Israel
{david.shavin, sagie.benaim}@mail.huji.ac.il

## Abstract

Vision Foundation Models (VFMs) have achieved remarkable success when applied to various downstream 2D tasks. Despite their effectiveness, they often exhibit a critical lack of 3D awareness. To this end, we introduce Splat and Distill, a framework that instills robust 3D awareness into 2D VFMs by augmenting the teacher model with a fast, feed-forward 3D reconstruction pipeline. Given 2D features produced by a teacher model, our method first lifts these features into an explicit 3D Gaussian representation, in a feedforward manner. These 3D features are then "splatted" onto novel viewpoints, producing a set of novel 2D feature maps used to supervise the student model, "distilling" geometrically grounded knowledge. By replacing slow per-scene optimization of prior work with our feed-forward lifting approach, our framework avoids feature-averaging artifacts, creating a dynamic learning process where the teacher's consistency improves alongside that of the student. We conduct a comprehensive evaluation on a suite of downstream tasks, including monocular depth estimation, surface normal estimation, multi-view correspondence, and semantic segmentation. Our method significantly outperforms prior works, not only achieving substantial gains in 3D awareness but also enhancing the underlying semantic richness of 2D features. Our project page and code are available at https://davidshavin4.github.io/Splat-and-Distill/.

## 1 Introduction

Vision Foundation Models (VFMs) such as DINO Caron et al. (2021) and DINOv2 Oquab et al. (2023) have achieved remarkable success by leveraging vast unlabeled 2D datasets via a student-teacher self-distillation paradigm, yielding robust and generalizable features. These features enable state-of-the-art results across a diverse array of downstream tasks such as semantic segmentation. Despite these advances, the capabilities of VFMs remain limited for 3D-aware tasks, like depth estimation, surface-normal reconstruction, and feature correspondence. Our work, therefore, aims to enhance the 3D awareness of such vision foundation models.

While several works focus on distilling 2D features into 3D representations, FiT3D Yue et al. (2024) takes the opposite approach: instilling 3D awareness into 2D VFMs by first lifting inconsistent 2D features into explicit 3D representations via per-scene optimization, then rendering views to create a dataset of "consistent" 2D features for fine-tuning. This method is fundamentally limited, as input features from different views are inconsistent El Banani et al. (2024), resulting in a *least-squares* compromise across views. You et al. (2024) uses a different approach, enforcing multi-view feature consistency through correspondences, bypassing explicit reconstruction. While this improves correspondence understanding, its supervision relies on enforcing feature similarity at corresponding points, which is insufficient for instilling the dense geometric understanding needed for complex downstream tasks.

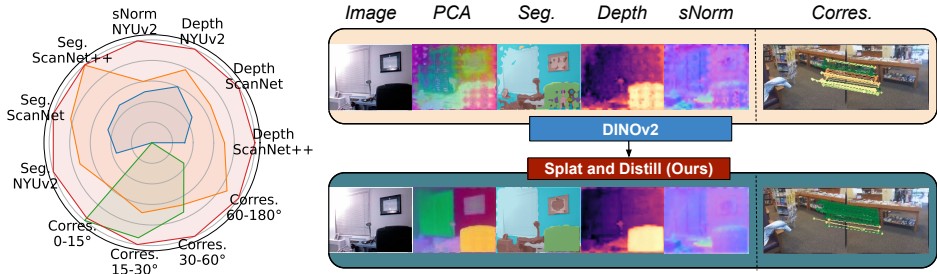

Figure 1: **Splat and Distill (SnD)** is a student-teacher distillation framework that augments the teacher with a feed-forward 3D reconstruction pipeline during training, resulting in 3D-aware 2D features. **Left:** Leveraging our approach on DINOv2, results in 2D features that enable state-of-the-art performance on downstream tasks such as monocular depth estimation (Depth), surface normal estimation (sNorm), semantic segmentation (Seg), and multiview correspondence (Corres). Shown here is comparison of SnD (our method) to vanilla DINOv2, and state-of-the-art approaches for improving 3D awareness,Fit3D (Yue et al., 2024), and MEF (You et al., 2024), based on a DINOv2 VIT-Small model, and considering the NYUv2 Silberman et al. (2012), ScanNet Dai et al. (2017) and ScanNet++ Yeshwanth et al. (2023) datasets (see further results in Sec. 4). For visualization, we provide normalized scores (min–max per metric, weakest baseline set to 0), using inverse RMSE for depth and normal estimation, IoU for segmentation, and Recall for correspondence (higher is better). See additional details in Sec. 4. **Right:** Visualization of our method compared to DINOv2.

To this end, we propose a fast, scalable alternative that avoids the inefficiencies and inconsistencies of optimization-based pipelines Yue et al. (2024), enabling complete, dense geometric scene understanding. Our key insight is that 3D consistency can be enforced by directly augmenting the teacher's architecture in a student-teacher paradigm. We initialize both the student and teacher models using a VFM that we aim to enhance with improved 3D awareness, along with a pre-trained 3D feed-forward reconstruction model, specifically DINOv2 for the former and MVSplat Chen et al. (2024) for the latter. We reconstruct scene appearance from a few context views, and lift semantics into it by extracting 2D feature maps from the context views using the teacher, upscaling them with segmentation masks, and attaching them to 3D via pixel-to-Gaussian correspondences. This allows efficient lifting of 2D features into 3D by attaching each feature to its corresponding Gaussian, avoiding slow per-scene optimization as in (Yue et al., 2024). We render features from the 3D scene at novel viewpoints and blend them with semantic masks, producing 2D feature maps as supervision for the student model. The student extracts features from these target views and learns to match the teacher's rendered augmented feature maps via gradient descent. The teacher and the student share the same architecture; the teacher's weights are updated using the exponential moving average (EMA) of the student's parameters, following the distillation objective of DINOv2.

This design confers numerous advantages over previous work. First, by iteratively adapting the features fed into the 3D reconstruction model via EMA, our method avoids the static "averaging" of inconsistent features that occurs in optimization-based approaches. Second, our framework learns a generalizable model for enforcing 3D consistency from a multitude of diverse scenes. Finally, by replacing this costly optimization with a feed-forward lifting mechanism, our method is significantly faster, more efficient, and more scalable, using much fewer Gaussians than previous work.

To demonstrate the efficacy of our approach, we conduct a comprehensive evaluation on a suite of downstream tasks. Following established protocols, we probe for 3D awareness through monocular depth estimation and surface normal prediction, which measures 3D awareness via a single image, as well as zero-shot multi-view feature correspondence to measure multi-view consistency. To ensure that these geometric gains do not come at the cost of semantic richness, we also evaluate performance on semantic segmentation. Our method significantly improves on the entire suite of tasks in comparison to state-of-the-art baselines, enabling enhanced single-view and multi-view 3D consistency as well as greater semantic richness. An illustration is provided in Fig. 1.

## 2 RELATED WORK

**Vision Foundation Models (VFMs).** Recent advances in ViT-based VFMs Dosovitskiy et al. (2020) have produced highly transferable visual representations that excel in a variety of 2D

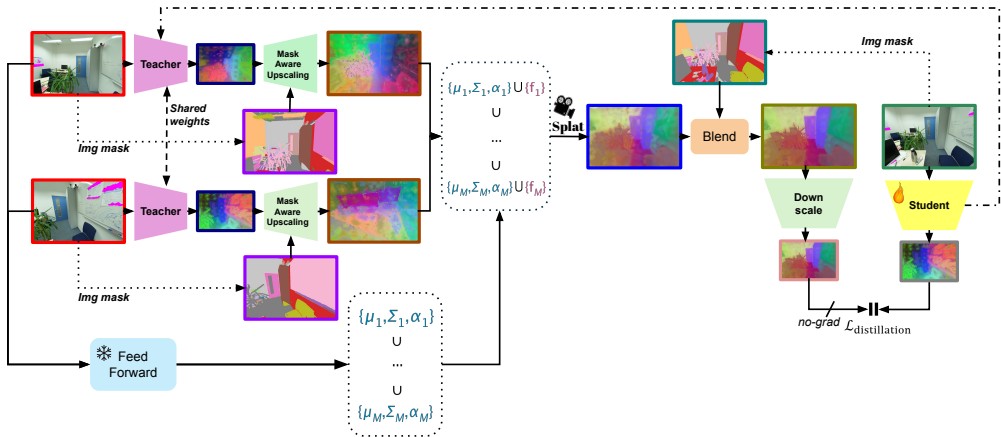

Figure 2: **Method Overview.** Starting from the LHS, two context views $\mathbf{I}_j^{ctx}$ are passed through a teacher network, producing two low-resolution 2D feature maps $\mathbf{F}_j^{ctx}$. Using corresponding semantic masks, mask-aware upscaling (Sec. 3.3) produces 2D features $\mathbf{F}_j^{high}$ of the input resolution. In parallel, a pretrained feed-forward 3D reconstruction model predicts 3D Gaussian primitives $\{\mu_j, \Sigma_j, \alpha_j\}$ using the same context views $\mathbf{I}_j^{ctx}$ (Sec. 3.2). The upscaled 2D feature maps, $\mathbf{F}_j^{high}$, are then lifted to these 3D Gaussian primitives, using 2D-3D correspondences, yielding a feature-augmented GS scene $\mathcal{G}_j \leftarrow \{\mu_j, \Sigma_j, \alpha_j\} \cup \{\mathbf{f}_j\}$ (Sec. 3.3). Next, the scene is splatted to a target viewpoint, producing a 2D feature map, which is then blended with the semantic mask of the target view, resulting in 2D features $\mathbf{F}_{blend}^{tgt}$ (Sec. 3.4). Concurrently, as shown on the RHS, the target image $\mathbf{I}^{tgt}$ (corresponding to the rendered viewpoint) is passed through the student network to obtain its feature map $\mathbf{F}_s^{tgt}$. $\mathbf{F}_{blend}^{tgt}$ is then downscaled (bilinearly) producing a lower resolution 2D feature map which is compared to $\mathbf{F}_s^{tgt}$ to supervise the student via a distillation loss (Sec. 3.5). The teacher's weights are updated as an EMA of the student's weights. *Note that SnD is finetuned on ScanNet++.*

tasks (Radford et al., 2021; Zhou et al., 2021; Touvron et al., 2022; He et al., 2022). Our work builds upon DINO and DINOv2 Caron et al. (2021); Oquab et al. (2023), which are based on a student-teacher self-distillation framework. In this framework, a student network learns to match the teacher's representations in different augmentations of the same image, resulting in embeddings with excellent performance on downstream tasks. Despite their success, recent work has shown that they remain limited in effectiveness in the 3D domain (Wang et al., 2022; Huang et al., 2024).

**3D Scene Representations.** Neural Radiance Fields (NeRF) (Mildenhall et al., 2021) have become a cornerstone for photorealistic Novel View Synthesis (NVS), but are hampered by slow rendering speeds. Recently, 3D Gaussian Splatting (3DGS) (Kerbl et al., 2023) has emerged as an explicit scene representation offering significantly faster rendering, while also achieving strong results in NVS. While providing high-quality 3D representations, these methods require per-scene optimization and numerous context views, making them unsuitable for our task.

**Feed-forward 3D Reconstruction.** To address these limits, recent methods use feed-forward pipelines to directly predict volumetric fields Chen et al. (2021); Yu et al. (2021) or 3D Gaussians Charatan et al. (2024); Wewer et al. (2024); Chen et al. (2024); Szymanowicz et al. (2024) from images. For instance, PixelSplat Charatan et al. (2024) predicts 3D Gaussians using cross-view attention and a Gaussian head, while MVSplat Chen et al. (2024) adds a cost-volume geometry encoder. These works focus on photometric reconstruction, lacking semantics and high-level features.

**3D Feature Distillation.** Building on these representations, several methods lift 2D features into 3D, enabling open-vocabulary understanding and editing capabilities (Kerr et al., 2023; Zhou et al., 2024; Qin et al., 2024; Labe et al., 2024; Levy et al., 2025; Marrie et al., 2025). Most pipelines are optimization-based, requiring slow per-scene fitting to align features with geometry. These methods focus on distilling 2D features into a 3D representation. By contrast, we distill 3D knowledge into 2D features using 3D representations as teachers, enhancing the 3D awareness of 2D features.

**Enhancing 3D Awareness Of VFMs.** A complementary line of research seeks to enhance the 3D awareness of pretrained 2D VFMs (Caron et al., 2021; Oquab et al., 2023). FiT3D Yue et al. (2024)

lifts 2D features into a 3D scene via optimization (Zhou et al., 2024), renders them from multiple views, and fine-tunes the 2D VFM by supervising its feature maps with rendered features. However, as input features from different views are inconsistent, optimization inevitably yields a *least-squares* compromise—a semantic blur averaging the initial errors. MEF You et al. (2024) improves 3D correspondence by enforcing feature similarity at corresponding points, but this relational constraint alone cannot instill the full, dense geometric scene understanding needed for complex downstream tasks. Our approach instead introduces 3D awareness within a distillation-based training by augmenting teachers with a feed-forward 3D reconstruction model.

## 3 METHOD

We now outline our approach, illustrated in Fig. 2. Finally, Sec.3.1 provides our student-teacher distillation approach of our method. Sec.3.2 describes the feed-forward 3D reconstruction pipeline augmenting the teacher. Sec.3.3 explains the process of lifting features into 3D, and Sec.3.4 details the mask-aware feature blending mechanism addressing sparse, irregular viewpoints. Sec.3.5 presents the overall loss formulation. Training and implementation details are in Appendix A.4.

### 3.1 STUDENT-TEACHER DISTILLATION FRAMEWORK

Our method is built upon the student-teacher self-distillation paradigm popularized by DINO and DINOv2 (Caron et al., 2021; Oquab et al., 2023). The architecture consists of a **student network**, $f_s$, with parameters $\theta_s$, and a **teacher network**, $f_t$, with parameters $\theta_t$, which share an identical network structure. A key departure from prior work is our supervisory mechanism. Instead of using 2D data augmentations, we leverage multi-view 3D scene data to instill geometric awareness. Our training data consists of scenes, where each scene $\mathcal{S}$ is a collection of images and their corresponding camera parameters. That is, $\mathcal{S} = \{(\mathbf{I}_i, \mathbf{P}_i)\}_{i=1}^N$, where $\mathbf{I}_i \in \mathbb{R}^{H \times W \times 3}$ is a given view and $\mathbf{P}_i \in \mathbb{R}^{3 \times 4}$ is the corresponding camera projection matrix. For each training iteration, we sample a scene and draw from it a pair of *context views*, $\{(\mathbf{I}_j^{\mathrm{ctx}}, \mathbf{P}_j^{\mathrm{ctx}})\}_{j=1}^2$, and a distinct *target view*, $(\mathbf{I}^{\mathrm{tgt}}, \mathbf{P}^{\mathrm{tgt}})$.

**3D-Aware Teacher Augmentation.** The core of our method lies in augmenting the teacher's output to be 3D-aware. This is achieved by generating a supervisory feature map for the target view through a 3D reconstruction and rendering pipeline. First, we use a pre-trained, feed-forward 3D reconstruction model (Sec . 3.2) to generate an explicit 3D representation of the scene, modeled as a set of 3D Gaussians, $\mathcal{G}_{\mathrm{geom}}$, from the two context views. Concurrently, we process the same context views with the teacher network $f_t$ to extract 2D feature maps, $\{\mathbf{F}_j^{\mathrm{ctx}} \in \mathbb{R}^{h \times w \times C}\}_{j=1}^2$, where $h \times w$ is the feature maps' spatial resolution. These features are then upscaled to $H \times W$ and *lifted* into 3D space by associating them with 3D Gaussians, yielding a 3D feature scene $\mathcal{G}_{\mathrm{feat}}$ (Sec. 3.3). Finally, this 3D feature scene is rendered from the perspective of the target view's camera $\mathbf{P}^{\mathrm{tgt}}$, producing the teacher's supervisory feature map, $\mathbf{F}_t^{\mathrm{tgt}} \in \mathbb{R}^{H \times W \times C}$ which goes through an additional blending mechanism to further enhance feature map quality (Sec. 3.4).

**Student Distillation.** The student network $f_s$ only observes the 2D target image $\mathbf{I}^{\mathrm{tgt}}$ and produces its own feature map, $\mathbf{F}_s^{\mathrm{tgt}} \in \mathbb{R}^{h \times w \times C}$. The student is trained by minimizing the discrepancy between its features and the teacher's rendered features, using the distillation loss described in Sec. 3.5.

### 3.2 FEED-FORWARD 3D RECONSTRUCTION MODEL

To provide a geometric scaffold for lifting 2D features into 3D, our method employs a pre-trained, feed-forward 3D reconstruction model based on the 3DGS representation (Kerbl et al., 2023). 3DGS models a scene as a collection of anisotropic 3D Gaussians, where each Gaussian $\mathcal{G}_i$ is parameterized by its geometric and appearance properties: a mean position $\mu_i \in \mathbb{R}^3$, a covariance matrix $\Sigma_i \in \mathbb{R}^{3 \times 3}$ (decomposed into scale and rotation), an opacity $\alpha_i \in [0, 1]$, and spherical harmonic (SH) coefficients $\mathbf{c}_i$ for view-dependent color.

Instead of traditional per-scene optimization, we leverage a feed-forward network, $\Phi_{\mathrm{geom}}$, which directly predicts the 3DGS representation from a sparse set of $K$ context views. Formally, given the context views $\{(\mathbf{I}_j^{\mathrm{ctx}}, \mathbf{P}_j^{\mathrm{ctx}})\}_{j=1}^K$ (we use $K = 2$), the model produces a set of $M$ Gaussians that represent the scene's geometry and appearance:

$$\Phi_{\mathrm{geom}} : \{(\mathbf{I}_j^{\mathrm{ctx}}, \mathbf{P}_j^{\mathrm{ctx}})\}_{j=1}^K \longmapsto \{\mathcal{G}_i\}_{i=1}^M \tag{1}$$

Specifically, we instantiate $\Phi_{\text{geom}}$ with a pre-trained MVSplat model (Chen et al., 2024). This model first extracts multi-view features, builds a cost volume to estimate per-pixel depth via plane-sweep stereo, and finally unprojects these depth maps to form the 3D Gaussian centers, while other Gaussian parameters are extracted from multi-view features using a Gaussian head. This process provides an explicit one-to-one correspondence between pixels in the context views and 3D Gaussians.

MVSplat is used as a frozen, off-the-shelf component. Since our objective is to construct a 3D *feature* scene rather than to perform novel view synthesis, we only utilize the geometric parameters $(\mu_i, \Sigma_i, \alpha_i)$ of the predicted Gaussians. The appearance parameters (the SH coefficients $\mathbf{c}_i$) are disregarded in the subsequent feature lifting step, which is detailed in Sec. 3.3.

## 3.3 Mask-Aware Feature Lifting

Having constructed a 3D geometric scaffold from the context views, the next step is to lift the 2D semantic features from the teacher network, $f_t$, onto this 3D representation. As noted above, this is achieved by processing the same context views $\{\mathbf{I}_j^{\text{ctx}}\}_{j=1}^2$ with the teacher to produce low-resolution feature maps $\{\mathbf{F}_j^{\text{ctx}} \in \mathbb{R}^{h \times w \times C}\}_{j=1}^2$. We then associate these features with the 3D Gaussians via the pixel-to-Gaussian correspondence provided by the reconstruction model.

A key challenge is the significant resolution mismatch between the teacher's patch-based feature maps ($h \times w$) and the full-resolution context images ($H \times W$) from which the Gaussians were derived (a scale of $\times 14$). Naively upscaling the feature maps using bilinear interpolation leads to severe blurring and feature mixing across object boundaries, degrading the final quality.

To address this, we propose to utilize a mask-aware upscaling mechanism that leverages instance semantic segmentation masks (available during training) to guide the interpolation. For each pixel $u$ in the target high-resolution grid, its feature value $\mathbf{F}_u^{\text{high}}$ is computed by interpolating only from neighboring low-resolution feature points $v$ that share the same semantic label. The interpolated feature is:

$$\mathbf{F}_u^{\text{high}} = \sum_{v \in \mathcal{N}(u)} w_{uv} \cdot \mathbf{F}_v^{\text{low}}, \tag{2}$$

where $\mathcal{N}(u)$ is the set of neighboring low-resolution feature points, and weights $w_{uv}$ are defined as:

$$w_{uv} = \begin{cases} \dfrac{\tilde{w}_{uv}}{\displaystyle\sum_{v' \in \mathcal{N}(u) \wedge \text{mask}(v') = \text{mask}(u)} \tilde{w}_{uv'}} & \text{if } \text{mask}(v) = \text{mask}(u), \\ 0 & \text{otherwise,} \end{cases} \tag{3}$$

where $\tilde{w}_{uv}$ are the standard bilinear interpolation weights and $\text{mask}(u)$ is the semantic label of pixel $u$. This formulation ensures that feature upscaling respects semantic boundaries, producing sharp, high-resolution feature maps $\mathbf{F}^{\text{high}} \in \mathbb{R}^{H \times W \times C}$. We demonstrate a quantitative and qualitative analysis of mask-aware upscaling in Sec. 4, and Fig.11 respectively. Our mask-aware lifting strategy is inspired by the feature lifting approach employed in OccamLGS (Cheng et al., 2024), utilizing semantic masks to guide interpolation and preserve object boundaries.

Finally, using the pixel-to-Gaussian correspondence from $\Phi_{\text{geom}}$, each feature vector in $\mathbf{F}^{\text{high}}$ is attached to its corresponding 3D Gaussian, resulting in a 3D feature scene where each Gaussian $\mathcal{G}_j$ is now endowed with a semantic feature vector $\mathbf{f}_j$: $\mathcal{G}_j \leftarrow \{\mu_j, \Sigma_j, \alpha_j\} \cup \{\mathbf{f}_j\}$.

## 3.4 Semantic Blending for Feature Regularization

Building a 3D scene from sparse and irregularly spaced context views can introduce geometric artifacts. When the 3D feature scene is rendered to a novel target view, these artifacts may manifest as noise or minor misalignments in the resulting feature map, $\mathbf{F}_{\text{rendered}}^{\text{tgt}}$. This noisy supervisory signal can degrade the quality of the student's learned representations.

To mitigate this, we introduce a semantic blending step that regularizes the rendered feature map by enforcing local consistency within object regions. Inspired by Huang et al. (2025), this step smooths the features spatially, guided by instance semantic segmentation masks. For each pixel location $u$ in the rendered feature map, the final blended feature $\mathbf{F}_{\text{blend}}(u)$ is computed as a weighted average of

the original rendered feature $\mathbf{F}_{\text{rendered}}(u)$ and the mean of all rendered features in $\mathcal{M}_u$, where $\mathcal{M}_u$ denotes the set of all coordinates in the target view sharing the same semantic mask as $u$.

$$\mathbf{F}_{\text{blend}}(u) = \alpha \cdot \mathbf{F}_{\text{rendered}}(u) + (1-\alpha) \cdot \frac{1}{|\mathcal{M}_u|} \sum_{v \in \mathcal{M}_u} \mathbf{F}_{\text{rendered}}(v), \tag{4}$$

where $\alpha \in [0,1]$ is a blending factor (we use $\alpha = 0.5$). By confining the averaging to within semantic boundaries, this process corrects for small geometric inconsistencies and produces a more coherent supervisory signal, while preserving sharp details at object edges. Visualization of this effect can be seen in the Appendix Fig. 12.

## 3.5 DISTILLATION OBJECTIVE

The final step is to distill the 3D-aware knowledge from the teacher into the student network. The supervisory signal is the blended feature map, $\mathbf{F}_{\text{blend}}^{\text{tgt}} \in \mathbb{R}^{H \times W \times C}$, which is the result of the full teacher pipeline noted above. Concurrently, the student network, $f_s$, processes the corresponding 2D target view $\mathbf{I}^{\text{tgt}}$ to produce its own feature map, $\mathbf{F}_s^{\text{tgt}} \in \mathbb{R}^{h \times w \times C}$. We first downscale the teacher's high-resolution feature map to match the student's output dimensions using bilinear interpolation.

Following the DINO framework (Caron et al., 2021; Oquab et al., 2023), both feature maps are passed through a shared DINO head which consists of a small MLP. The student's parameters $\theta_s$ are optimized to minimize a distillation loss $\mathcal{L}_{\text{distill}}$:.

$$\min_{\theta_s} \mathcal{L}_{\text{distill}}(\text{head}(\mathbf{F}_s^{\text{tgt}}), \text{sg}(\text{head}(\mathbf{F}_{blend}^{\text{tgt}}))) \tag{5}$$

where $\text{head}(\cdot)$ is the DINO head, $\mathbf{F}_{blend}^{\text{tgt}}$ is the rendered teacher features after blending and down-scaling, $\text{sg}(\cdot)$ is the stop-gradient operator, and $\mathcal{L}_{\text{distill}}$ is the cross-entropy loss between the teacher and student's output distributions. The student's parameters $\theta_s$ are optimized via backpropagation, while the teacher's parameters are instead updated via EMA: $\theta_t \leftarrow \lambda\theta_t + (1-\lambda)\theta_s$, where $\lambda \in [0,1]$ is the momentum coefficient. More details are in Appendix A.4.

## 4 EXPERIMENTS

We evaluate our ability to enhance the 3D awareness of DINOv2 features while improving their semantic representation. Specifically, we assess the inference of 3D perceptual properties (surface normal and depth estimation) from single-image features and multi-view feature correspondence. For semantics, we evaluate semantic segmentation. Limitations are provided in Appendix A.8.

**Baselines.** Our first baseline is the vanilla pre-trained DINOv2 model. We conduct our experiments on the small or base variants. We also consider Fit3D Yue et al. (2024), the work most closely related to ours. Consequently, we conducted our finetuning (on pretrained DINOv2 student and teacher) using the same subset of ScanNet++ data Yeshwanth et al. (2023) as is done by Fit3D, to ensure a fair comparison. Fit3D first constructs a dataset of 3DGS scenes with DINOv2 features by following established feature distillation approaches. It then fine-tunes DINOv2 to produce features that match those rendered from the optimized 3D scenes. Lastly, we consider MEF You et al. (2024), which fine-tunes a DINOv2 by enforcing multiview feature correspondence in their training objective. However, we note that MEF requires correspondence annotation, which our method does not use. Implementation details for evaluating the different baselines can be found in Appendix A.6.

**Evaluation Protocol.** For monocular depth estimation, surface normal estimation, and semantic segmentation, we consider the *linear probing* protocol. We also evaluate multi-view consistency by measuring the correspondence between multiple views, where the goal is to identify image patches across views that depict the same 3D point. See evaluation details in Appendix A.5.

**Evaluation Datasets.** For depth estimation and semantic segmentation, we follow Fit3D and consider indoor scenes from the ScanNet++ validation set Yeshwanth et al. (2023) as well as ScanNet Dai et al. (2017) and NYUv2 Silberman et al. (2012) datasets. These datasets share similar characteristics but employ different sensor modalities. For surface normal estimation, we use the NYUv2 dataset where surface normal annotation is curated by GeoNet Ladický et al. (2014). For feature correspondence, we evaluate on the test set of SuperGlue Sarlin et al. (2020),

Table 1: Quantitative comparison for **monocular depth estimation**, on ViT-Small/Base backbones.

| | Method | ScanNet++ | | ScanNet | | NYUv2 | |
|---|---|---|---|---|---|---|---|
| | | Rel ↓ | RMSE ↓ | Rel ↓ | RMSE ↓ | Rel ↓ | RMSE ↓ |
| **ViTs** | DINOv2 | 0.2811 | 0.3777 | 0.1437 | 0.2817 | 0.1476 | 0.5210 |
| | Fit3D | 0.2500 | 0.3506 | 0.1375 | 0.2713 | 0.1418 | 0.5075 |
| | MEF | 0.3085 | 0.4000 | 0.1566 | 0.3042 | 0.1661 | 0.5656 |
| | Ours | **0.2421** | **0.3299** | **0.1266** | **0.2555** | **0.1406** | **0.4912** |
| **ViTb** | DINOv2 | 0.2539 | 0.3435 | 0.1169 | 0.2369 | 0.1375 | 0.4948 |
| | Fit3D | 0.2420 | 0.3306 | 0.1166 | 0.2346 | 0.1359 | 0.4794 |
| | MEF | 0.2849 | 0.3726 | 0.1269 | 0.2534 | 0.1537 | 0.5214 |
| | Ours | **0.2169** | **0.2971** | **0.1113** | **0.2245** | **0.1261** | **0.4596** |

| | Image | GT | DINOv2 | MEF | Fit3D | Ours |
|---|---|---|---|---|---|---|

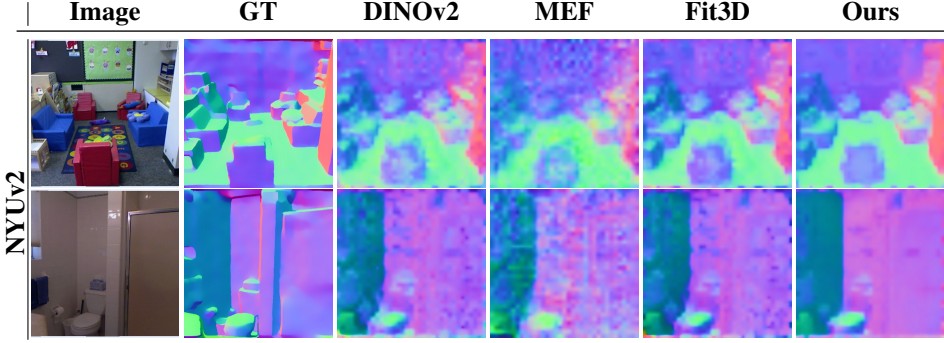

Figure 3: Qualitative comparison for **monocular depth estimation** using ViT-Small backbone (GT=Ground Truth).

| Image | GT | DINOv2 | MEF | Fit3D | Ours |
|---|---|---|---|---|---|

Figure 4: Qualitative comparison of **surface normals estimation** using ViT-Small backbone.

which includes image pairs from ScanNet. To further assess robustness and transferability, we include OOD benchmarks: ADE20k Zhou et al. (2017) and Pascal VOC Everingham et al. (2015) for semantic segmentation, and the KITTI Geiger et al. (2013) dataset for depth estimation. For further information on datasets and specific training-test splits employed, please refer to Appendix A.7.

### 4.1 IN-DOMAIN EVALUATION

We evaluate our features on downstream tasks for in-domain datasets.

**Monocular Depth Estimation.** Tab.1 shows quantitative results for monocular depth estimation using RMSE and Abs-Rel metrics. Our method consistently outperforms baselines, with average relative gains on RMSE of **5.90%**, **5.82%**, and **3.21%** on ScanNet++, ScanNet, and NYUv2 over the closest baseline. Fig. 3 presents a visual comparison of our method to baselines on the NYUv2 dataset, where results on the ScanNet and ScanNet++ are shown in Appendix A.2 Fig. 8. Our depth maps exhibit more refined structural details and smoother geometric surfaces than baselines.

Table 2: Quantitative comparison for **surface normal estimation** on ViT-Small/Base backbones.

| | Method | NYUv2 RMSE ↓ |
|---|---|---|
| **ViTs** | DINOv2 | 30.99 |
| | Fit3D | 30.57 |
| | MEF | 33.05 |
| | Ours | **28.93** |
| **ViTb** | DINOv2 | 31.40 |
| | Fit3D | 30.57 |
| | MEF | 32.60 |
| | Ours | **29.37** |

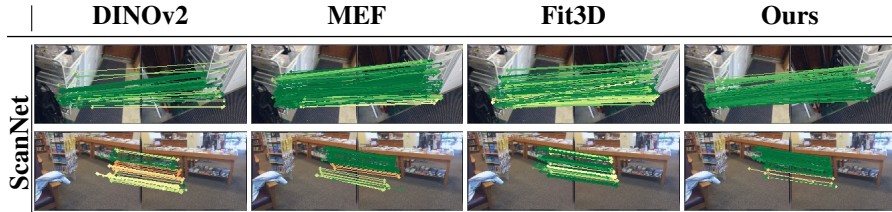

Figure 5: Qualitative comparison of **multi-view correspondences** using ViT-Small backbone. Lines connect matched points between the two views; color encodes the 2D Euclidean reprojection error computed under the ground-truth pose, with green/red indicating small/large error, respectively.

Table 3: Quantitative comparison for **semantic segmentation** using ViT-Small/Base.

|  | Method | ScanNet++ | | | ScanNet | | | NYUv2 | | |
|---|---|---|---|---|---|---|---|---|---|---|
|  |  | aAcc ↑ | mIoU ↑ | mAcc ↑ | aAcc ↑ | mIoU ↑ | mAcc ↑ | aAcc ↑ | mIoU ↑ | mAcc ↑ |
| ViTs | DINOv2 | 80.23 | 29.54 | 39.11 | 76.60 | 51.27 | 63.28 | 82.25 | 64.73 | 75.56 |
| | Fit3D | 83.34 | 31.77 | 41.09 | 78.53 | 54.50 | 66.57 | 83.32 | 66.33 | 77.06 |
| | MEF | 79.45 | 27.44 | 36.77 | 74.63 | 47.44 | 58.98 | 81.02 | 63.17 | 74.16 |
| | Ours | **83.84** | **31.78** | **41.42** | **79.48** | **56.01** | **68.10** | **84.31** | **67.50** | **77.96** |
| ViTb | DINOv2 | 81.85 | 31.95 | 41.69 | 79.48 | 56.42 | 68.22 | 83.92 | 67.47 | 78.02 |
| | Fit3D | **84.90** | **34.85** | **44.83** | 82.25 | 60.78 | 72.60 | 85.54 | 70.18 | 80.36 |
| | MEF | 82.23 | 31.63 | 41.50 | 78.60 | 54.14 | 65.83 | 83.62 | 67.64 | 78.12 |
| | Ours | 84.77 | 34.07 | 44.00 | **83.43** | **62.64** | **74.29** | **86.05** | **70.60** | **80.91** |

**Surface Normal Estimation.** In Tab. 2, we present the RMSE over dense prediction of normal directions on single-view images on the NYUv2 dataset. Our method yields an improvement of **5.37%** over the closest baseline using DINOv2-Small and **3.93%** using DINOv2-Base. Fig. 4 further highlights qualitatively the superior quality of our predictions. Our model produces smoother normal maps (e.g., second row) and demonstrates a more accurate understanding of the 3D scene. For example, when viewing the couch, other methods incorrectly predict normals, suggesting the couch faces the camera. Our model correctly infers the visible surface is the back of the couch and assigns flat, consistent normals.

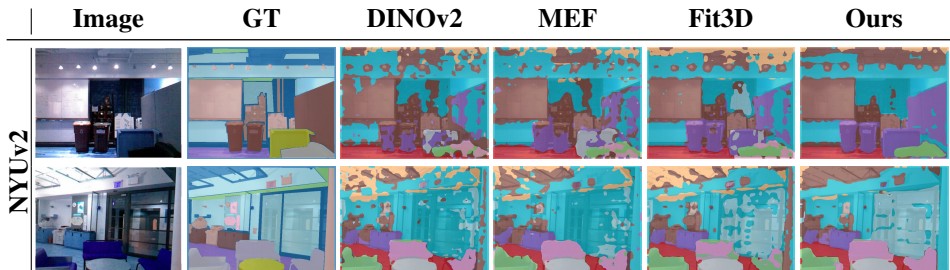

Figure 6: Qualitative comparison for **semantic segmentation** using ViT-Small backbone.

**Multiview Correspondence.** For multi-view correspondence, we fine-tune DINOv2 without semantic blending of the splatted features, as blending tends to smooth out feature representations. We consider the recall of the fraction of all proposed correspondences that satisfy an accuracy criterion of ten pixels, see Appendix A.5. In Fig. 7, we observe our method consistently improves recall over baselines across varying viewpoint changes, In Fig. 5, we illustrate this qualitatively.

**Semantic Segmentation.** In Tab. 3, we provide a quantitative comparison for semantic segmentation. We report average accuracy (aAcc), mean intersection over union (mIoU), and mean accuracy (mAcc). Compared to the leading baseline of FiT3D when using

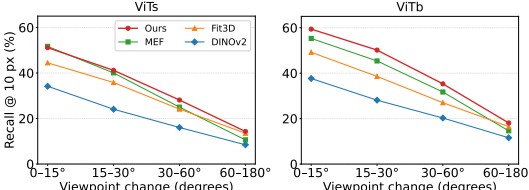

Figure 7: Quantitative comparison of **multi-view correspondence** for varying viewpoint changes, using ViT-Small/Base backbones on ScanNet.

Table 4: Quantitative comparison **on out-of-domain datasets**, using a ViT-Base backbone.

| Method | ADE20k (seg.) | | | Pascal VOC (seg.) | | | KITTI (depth) | |
|---|---|---|---|---|---|---|---|---|
| | aAcc ↑ | mIoU ↑ | mAcc ↑ | aAcc ↑ | mIoU ↑ | mAcc ↑ | Rel ↓ | RMSE ↓ |
| DINOv2 | 77.39 | 40.78 | 53.31 | 96.20 | 83.00 | 89.36 | 0.0686 | 2.3558 |
| Fit3D | 82.21 | 48.29 | 60.03 | 96.77 | 85.08 | 90.72 | 0.0679 | 2.2485 |
| MEF | 80.16 | 45.14 | 56.15 | 95.73 | 80.82 | 87.67 | 0.0772 | 2.4160 |
| Ours | **83.24** | **50.01** | **61.61** | **96.96** | **85.75** | **91.63** | **0.0631** | **2.1741** |

Table 5: **Ablation studies**, on ScanNet++ dataset, using the VIT-Small variant.

| Ablation | Segmentation | | | Depth | |
|---|---|---|---|---|---|
| | aAcc↑ | mIoU↑ | mAcc↑ | Rel↓ | RMSE↓ |
| Without Blending (A) | 82.83 | 30.99 | 40.80 | 0.2531 | 0.3435 |
| Bilinear instead of Masked Upscaling (B) | 83.66 | 31.46 | 41.01 | 0.2428 | 0.3309 |
| Cosine Loss instead of Distillation Loss (C) | 83.54 | 31.27 | 40.96 | 0.2421 | 0.3310 |
| Frozen instead of Learnable Teacher (D) | 83.34 | 31.90 | **41.88** | 0.2500 | 0.3444 |
| Context instead of Novel Views (E) | **84.02** | **32.08** | 41.74 | 0.2430 | 0.3332 |
| SAM Masks instead of Manual Masks (F) | 83.49 | 31.51 | 41.39 | 0.2436 | 0.3328 |
| Feature Rendering Loss (G) | 83.06 | 31.40 | 40.97 | 0.2484 | 0.3430 |
| Basic Variant (H) | 81.80 | 30.66 | 40.47 | 0.2741 | 0.3520 |
| Ours (Full Model) | 83.84 | 31.78 | 41.42 | **0.2421** | **0.3299** |

DINOv2-Small backbone, we achieve a relative improvement of **0.03%**, **2.77%**, and **1.76%** on ScanNet++, ScanNet, and NYUv2, considering mIoU.

We observe a similar trend with DINOv2-Base, improving significantly on the ScanNet and NYUv2 datasets, with a slight decrease on the ScanNet++ dataset. In Fig. 6, we present a qualitative comparison of on the NYUv2 dataset. Our method produces cleaner object boundaries and avoids the fragmented masks present in baselines, as seen in the partition wall in the first row. Additional visualizations, for additional datasets, can be seen in Appendix A.2 in Fig. 9.

## 4.2 OUT OF DOMAIN EVALUATION

In Tab. 4, we assess our method using the DINOv2-base backbone on out-of-domain datasets. For segmentation on ADE20K and Pascal VOC, we achieve relative mIoU improvements of **3.56%** and **0.79%**, respectively over closest baseline. Notably, ADE20K contains many outdoor, highly cluttered scenes distinct from our training data. For monocular depth estimation on KITTI Geiger et al. (2013), we achieve a **3.31%** improvement over Fit3D, demonstrating transfer of 3D spatial awareness from indoor to outdoor scenarios. See Appendix A.2 Fig. 10 for visualizations.

## 4.3 ABLATION STUDIES AND FEATURE VISUALIZATION

In Tab. 5, we ablate our model on semantic segmentation and depth estimation. In **Ablation A**, we consider the effect of removing blending (Sec. 3.4). We also provide a visual illustration in Appendix A.2 Fig. 12. Next, in **Ablation B**, we consider the effect of replacing the mask-aware upscaling with standard bilinear upscaling. Visually, the effect is shown in Appendix A.2 Fig. 11. In **Ablation C**, we consider the effect of our distillation objective and compare it to using a cosine loss on the patch embeddings (without DINO head) instead. In **Ablation D**, we consider the effect of freezing the teacher, as opposed to updating it using EMA, demonstrating the effectiveness of jointly updating the teacher and student. In **Ablation E**, we consider the effect of rendering features to context views instead of a target view located between the two context views. As seen, this is beneficial for depth estimation. Interestingly, segmentation performance improves when rendering to the context views. In **Ablation F**, we use SAM-extracted masks Kirillov et al. (2023) instead of manually annotated masks for *mask-aware upscaling* and *semantic blending*, showing that manual

annotation has only a minor advantage. Note, we don't require consistent class labels across frames. In **Ablation G**, we replace our student-teacher distillation framework with a direct feature rendering loss on a fixed teacher. While this improves over Vanilla DINOv2, it underperforms our Full SnD (Depth RMSE 0.3430 vs. 0.3299), confirming that our soft distillation objective and iterative teacher updates are essential for mitigating artifacts and maximizing geometric awareness. In **Ablation H**, we evaluate the most basic configuration of our method (Fixed teacher, no mask-aware upscaling, no blending). The performance drop relative to the full model (Depth RMSE 0.3520 vs. 0.3299) validates that our architectural components—specifically the iterative EMA update and mask-aware lifting—are integral to achieving state-of-the-art results.

**Feature Visualization.** In Appendix A.3, we visualize our features using PCA, further showcasing our approach's advantages. These reveal less noise and clearer semantic boundaries in feature space. K-means further shows that semantically similar objects cluster together while retaining fine details. We also analyze features from two views, in shared and per-view spaces.

## 5 CONCLUSION

We introduced *Splat and Distill*, a novel 3D-aware distillation framework to instill robust 3D awareness into 2D VFMs. Our core contribution is the augmentation of the teacher network with a fast, feed-forward 3D reconstruction pipeline within a distillation framework. This allows us to lift 2D features from context views into an explicit 3D Gaussian representation, *splat* these features onto novel viewpoints, and *distill* this geometrically grounded knowledge into a student model. Our method significantly outperforms state-of-the-art baselines on a comprehensive suite of downstream tasks, including monocular depth estimation, surface normal estimation, and multi-view correspondence, while also enhancing the underlying semantic richness of the original 2D features.

## REPRODUCIBILITY STATEMENT

We provide full implementation details of our method in Appendix A, including architecture specifications, hyperparameters, and training procedures. In addition, we include the source code as supplementary as part of the submission, ensuring full reproducibility of experiments. All datasets used in this work are publicly available online under their respective licenses, and we describe in Appendix A.7 how they are accessed and preprocessed.

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

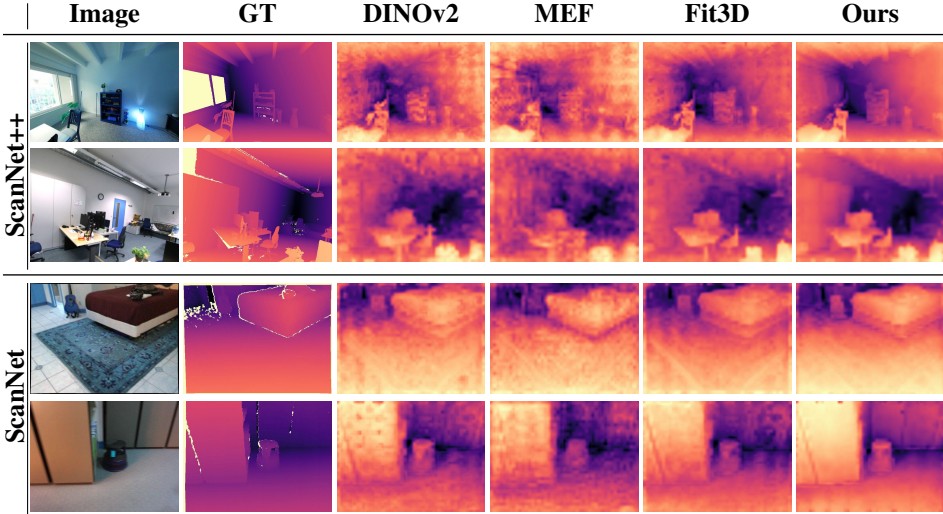

Figure 8: Qualitative comparison on the task of **monocular depth estimation**, using a ViT-Small backbone for the ScanNet++ and ScanNet datasets.

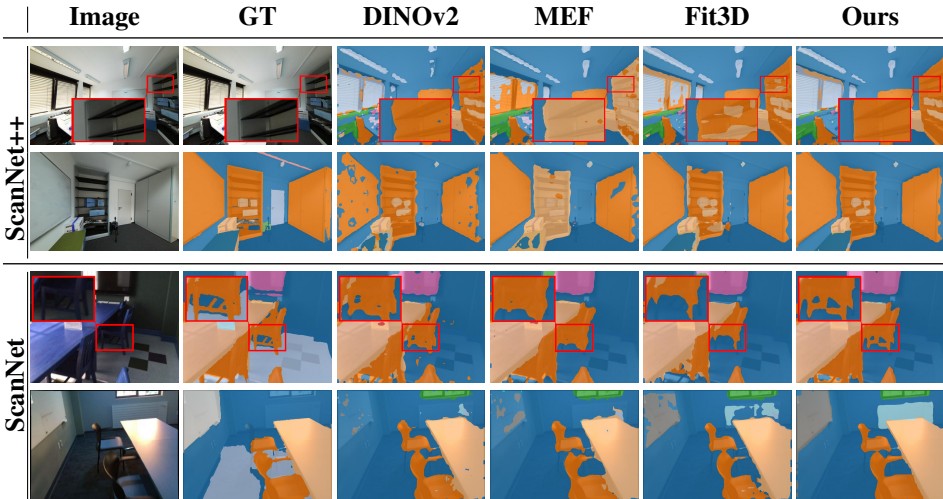

Figure 9: Qualitative comparison on the task of **semantic segmentation**, using a ViT-Small backbone for the ScanNet++ and ScanNet datasets.

## A  APPENDIX

We present additional comparisons in Appendix A.1. Qualitative results are in Appendix A.2, and in Appendix A.3, we provide a qualitative analysis of feature representations, comparing our method to DINOv2. Training implementation details are described in Appendix A.4, and evaluation protocols for all downstream tasks are summarized in Appendix A.5. Appendix A.6 outlines the baseline evaluation procedures, while Appendix A.7 details all datasets used for training and validation. In Appendix A.8 we discuss the limitations of our method. Finally, Appendix A.9 clarifies the use of large language models for literature review and manuscript preparation.

### A.1  ADDITIONAL COMPARISONS: TASK-SPECIFIC BACKBONES AND CONCURRENT WORK

**Task-Specific Backbones (VGGT).** To evaluate SnD as a backbone within state-of-the-art pipelines, we integrated our model into VGGT (Wang et al., 2025) for monocular depth estimation on ScanNet++. Since the pre-trained VGGT relies on ViT-Large (1024-dim) and our model is ViT-Small

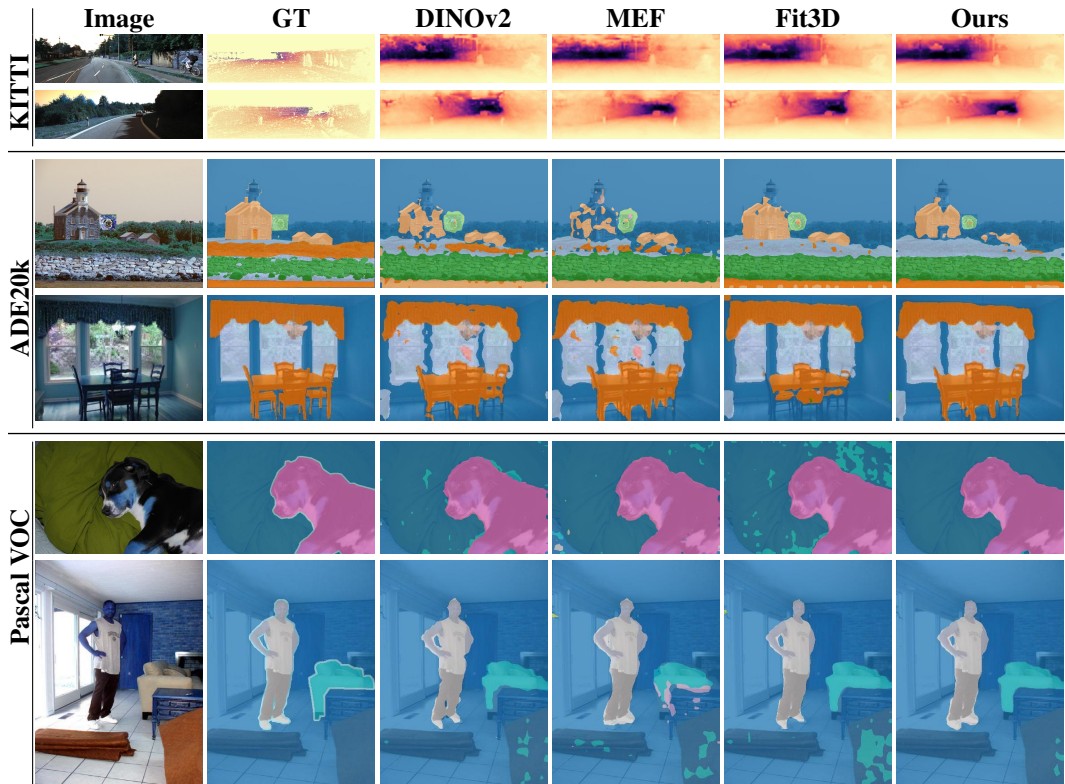

Figure 10: Qualitative comparison on **out-of-domain datasets** for two tasks: **monocular depth estimation and semantic segmentation**, using a ViT-Small backbone. Monocular depth estimation is evaluated on the KITTI dataset, while semantic segmentation on ADE20K and Pascal VOC.

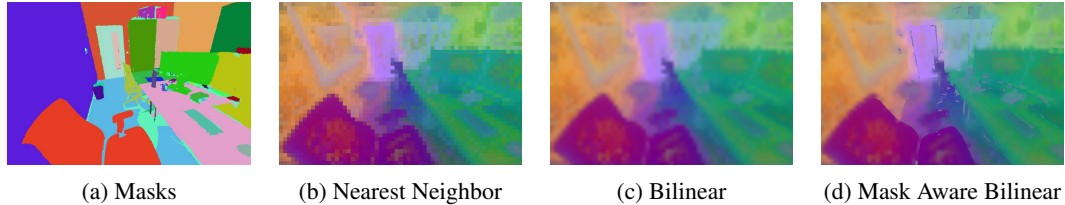

(a) Masks     (b) Nearest Neighbor     (c) Bilinear     (d) Mask Aware Bilinear

Figure 11: **Effect of mask-aware upscaling.** PCA of upscaled features with different methods, comparing nearest-neighbor vs. bilinear vs. mask-aware upscaling. Using mask-aware upscaling enables clear boundaries between distinct objects.

(384-dim), we added a learnable linear projection ($384 \rightarrow 1024$). Crucially, to isolate the quality of the backbone representations, we froze the entire VGGT architecture (transformer layers, global/frame attention) and fine-tuned only the projection and depth head. As shown in Tab. 6 (Left), replacing the DINOv2 backbone with SnD yields improved performance.

**Comparison to DUNE.** We compare against DUNE (Sarıyıldız et al., 2025), which distills a universal encoder. We evaluated the official DUNE weights on ScanNet++, specifically the `dune_vitsmall14_448.pth` variant as it produced better results than the 336px version. As shown in Tab. 6 (Right), SnD significantly outperforms DUNE on both depth estimation and semantic segmentation. Unlike DUNE, which inherits 3D inconsistencies from DINOv2, SnD corrects them via our feed-forward reconstruction pipeline.

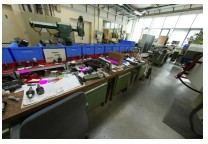 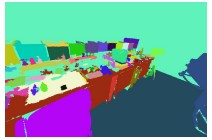 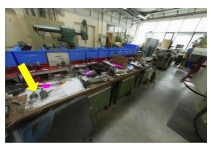 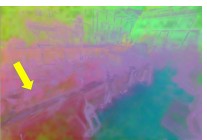 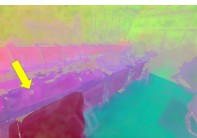

(a) View      (b) Mask      (c) Rendered Image      (d) Splatted Features      (e) Blended Features

Figure 12: **Semantic Blending for Feature Regularization Visualization.** We visualize the impact of semantic blending for feature regularization. (a) shows the input view, (b) displays its corresponding semantic mask, available during training. (c) presents the reconstructed scene generated by our pretrained feed-forward model, highlighting areas of poor reconstruction—such as the misaligned desk. The splatted features (visualized via PCA) are shown in (d), illustrating the inaccuracies introduced by flawed reconstruction. (e) demonstrates how semantic blending, guided by the mask, refines these features and produces sharper object boundaries.

Table 6: **Left:** Comparison of backbones within the VGGT pipeline (fine-tuning only the head/projection). **Right:** Comparison against DUNE. Both evaluated on ScanNet++ (ViT-Small).

| VGGT Backbone Comparison | | | Comparison vs. DUNE | | |
|---|---|---|---|---|---|
| Method | Depth Abs-Rel ↓ | Depth RMSE ↓ | Method | Depth RMSE ↓ | Seg. mIoU ↑ |
| VGGT (DINOv2) | 0.2220 | 0.3426 | DUNE | 0.3929 | 25.77 |
| **VGGT (SnD)** | **0.2117** | **0.3283** | **Ours (SnD)** | **0.3299** | **31.78** |

## A.2 ADDITIONAL QUALITATIVE RESULTS

We provide additional qualitative results for the ScanNet and ScanNet++ dataset, on single-view depth estimation, and semantic segmentation results on indoor scenes in Fig. 8, and Fig. 9, respectively. In addition, we visualize single-view depth estimation and semantic segmentation results on out-of-domain datasets in Fig. 10.

**Depth estimation qualitative results.** Fig. 8 presents qualitative results for depth estimation on the ScanNet++ and ScanNet datasets. Our method accurately captures fine details, such as the chair in the second row, and produces smoother surfaces, as observed on the bed in the third row. Depth estimation captures the global structure of a scene, while surface normal prediction provides local orientation cues. Although their performance is often correlated, these tasks rely on distinct visual signals and thus offer complementary evidence for 3D understanding (El Banani et al., 2024).

**Semantic segmentation qualitative results.** Figure 9 shows qualitative results for semantic segmentation on ScanNet++ and ScanNet. Our method produces sharper, more coherent masks; for instance, in the third row, it accurately segments chair legs in the dining table image, even under low-light conditions. These results indicate that our model learns representations with enhanced spatial and semantic consistency.

**Out-of-domain qualitative results.** Although our model was trained solely on the ScanNet++ dataset of indoor scenes, it demonstrates strong generalization to out-of-domain datasets. In Fig. 10, we evaluate our method on the KITTI dataset for depth estimation, observing improved depth smoothness. For semantic segmentation, comparisons on ADE20k and Pascal VOC show that our approach captures fine details, such as the legs of chairs at the dining table (fourth row), and produces cleaner, less noisy outputs, as seen in the background of the dog image (fifth row).

**Visualization of mask-aware upscaling.** We visualize the impact of various upscaling methods in comparison to the mask-aware bilinear upscaling (see Sec. 3.3) used by our method in Fig. 11. Notably, using mask-aware upscaling produces sharp boundaries between objects by leveraging semantic masks to guide interpolation around object boundaries.

**Visualization of semantic blending for feature regularization.** To analyze the impact of our semantic blending (see Sec. 3.4), we demonstrate its effect on rendered features visually, using

PCA. In Fig. 12, we showcase an example where blending corrects a poorly reconstructed scene. Notably, observe the initial poor reconstruction of the chair and its correction after the application of blending.

| **Image** | **DINOv2 PCA** | **Ours PCA** | **DINOv2 K-means** | **Ours K-means** |

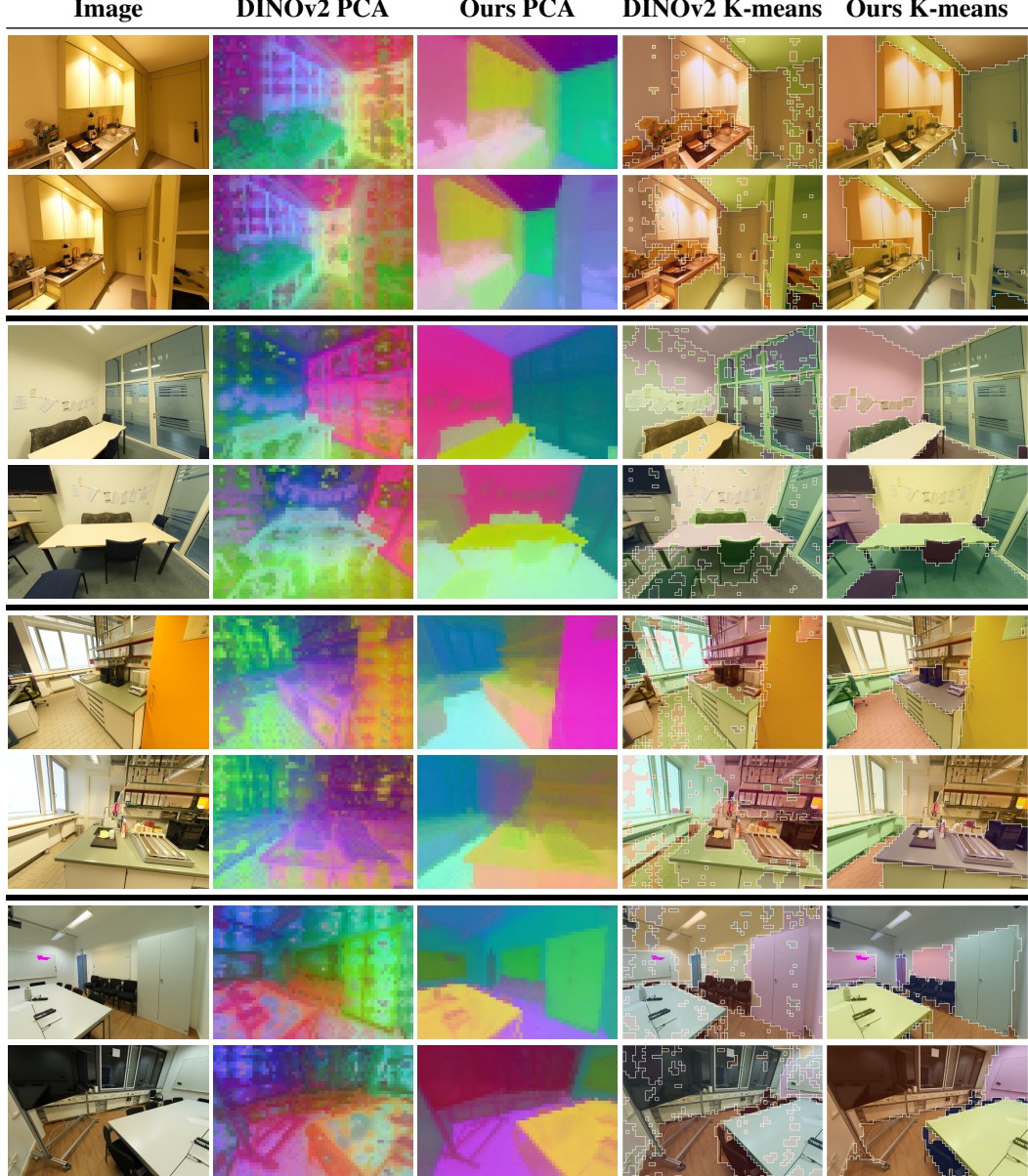

Figure 13: Qualitative visualization of DINOv2 features vs. ours on a ViT-Small backbone. Each pair of rows corresponds to a scene from the ScanNet++ dataset. PCA and K-means are computed per scene rather than per image.

### A.3 FEATURES QUALITATIVE ANALYSIS

*Multi-view features.* We provide a qualitative analysis of our feature representations using principal component analysis (PCA) and K-means clustering. In Fig.13, we compare results before and after applying our distillation method. To assess multi-view consistency, we extract features from two distinct views of the same scene from ScanNet++ (Yeshwanth et al., 2023). Both PCA and K-means clustering are performed jointly on features from both views, ensuring a shared feature basis across perspectives. Our results reveal clear and well-defined boundaries in the PCA plots, as well as more

compact clusters in the K-means visualizations. These findings indicate improvements in multi-view consistency and feature semantics.

*Single-view features.* In Fig.14 we evaluate feature quality across a diverse range of datasets, including both in-domain and out-of-domain samples. Our analysis reveals improved feature semantics, as evidenced by reduced noise in the PCA projections and more compact clusters in the K-means visualizations. Importantly, we find that distinct object semantics present before applying our distillation method are also preserved afterward, across all datasets (for example, in the seventh row, K-means consistently identifies *chickens* before and after distillation). These results increase the confidence that the proposed approach maintains the semantic integrity of the original model's features while improving its quality via 3D awareness.

### A.4 TRAINING IMPLEMENTATION DETAILS

For 3D scene reconstruction, we employ an off-the-shelf, pre-trained MVSplat model Chen et al. (2024), originally trained on the RE10k dataset (Zhou et al., 2018). We further fine-tune this model on ScanNet++ Yeshwanth et al. (2023) to improve its 3D reconstruction for 40,000 iterations to better adapt it to our specific task requirements.

We initialize both the *student* and the *teacher* with the pretrained DINOv2 weights available online. In addition, we require a DINO head Oquab et al. (2023). The DINO head maps each backbone feature to a probability distribution over a set of learnable prototypes, implemented as the weights of a final linear layer. These prototypes act as latent surrogate classes, turning the distillation task into a classification-like objective without labels. An EMA updated teacher network provides soft targets, and the student is trained to match them via cross-entropy. This design encourages features to organize around diverse prototype centroids, prevents collapse, and yields robust, transferable representations, as refined in DINOv2 with additional stabilization techniques. The DINO head design used in our experiments consists of a multi-layer perceptron (MLP) with 3 layers, where each hidden layer has a dimensionality of 2048. A bottleneck layer with a reduced dimension of 256 is placed before the output, which helps to control the capacity and regularize the representations. The final output projects features onto 65,536 prototype vectors, which serve as anchor points for the distillation objective. This architectural choice follows the standard DINO framework, facilitating scalable clustering and robust representation learning.

Fine-tuning of the DINOv2 Oquab et al. (2023) backbone is performed using LoRA Hu et al. (2022) with a rank of 8 to enable efficient adaptation while minimizing additional parameters. Optimization is carried out with a batch size of 1 using the Adam optimizer Kingma & Ba (2014), starting from an initial learning rate of $2 \times 10^{-5}$ and following a cosine annealing schedule.

The teacher network is updated every 10 training steps via EMA with a momentum coefficient of 0.999 (Tarvainen & Valpola, 2017). For simplicity, we do not incorporate DINOv2-specific components such as temperature scheduling and momentum centering.

Training is conducted for 50,000 iterations on a single NVIDIA L40S GPU, requiring approximately 18 hours to complete. The primary computational bottleneck of our approach arises from the trainable DINO head and the use of high-resolution input images. Note that the entire 3D reconstruction process and feature extraction from the teacher network are performed with frozen weights, making these steps highly efficient in resources.

### A.5 EVALUATION DETAILS

*Monocular Depth Estimation.* We treat depth estimation as a classification problem by discretizing the continuous depth range into 256 uniformly spaced bins, following the AdaBins approach Bhat et al. (2021). The input to the classification head is constructed by concatenating the global `[CLS]` token with each patch token from the Vision Transformer, and the resulting features are spatially upscaled by a factor of four. A linear head is then applied to these upscaled features to produce, for each pixel, logits corresponding to each depth bin. To map these logits to depth predictions, we generate (n_bins) linearly spaced depth values between (mindepth) and (maxdepth). The logits are transformed into a probability distribution across bins by applying a ReLU activation, adding a small epsilon for numerical stability, and normalizing so that the probabilities sum to one. The final depth for each pixel is calculated as the weighted sum of the bin centers, with the weights given

by the normalized probabilities. This produces a continuous depth map, which is then interpolated back to the spatial resolution of the original input image. The probing is done using a classification loss, for 307,200 iterations on a single GPU to predict the correct depth bin for each patch. We use AdamW optimizer Kingma & Ba (2014) with a learning rate of 0.0001, betas (0.9, 0.999), weight decay of 0.01, and a batch size of two.

*Surface Normal Estimation.* For surface normal estimation, we follow the protocol of Bae et al. Bae et al. (2021) and employ an uncertainty-aware angular loss. We train a linear head to regress surface normals from frozen features, which are upscaled by a factor of four. The output of the linear layer is subsequently interpolated to the original input image resolution. Training is performed for ten epochs with a batch size of eight, using the AdamW optimizer and a learning rate of $5 \times 10^{-4}$. The learning rate schedule consists of a linear warmup phase followed by cosine decay(Kingma & Ba, 2014; Loshchilov & Hutter, 2016). We report the root mean squared angular prediction error as our evaluation metric.

*Feature correspondence* We follow the protocol established by Probe3D El Banani et al. (2024). To find correspondences between two images, we first extract feature maps from each image using a neural network. Pixel coordinates are projected into 3D space using depth maps and camera intrinsics, and features at these 3D locations form point clouds. We then match features from the two point clouds by performing k-nearest neighbor search using cosine similarity. A ratio test is applied to filter out ambiguous matches, and the remaining matches are selected as correspondences. For evaluation, we use the ground-truth relative pose between the two views to transform the corresponding 3D points from one view to the other. These transformed points are then projected back to 2D image coordinates in the target image. We measure correspondence accuracy as the Euclidean distance between each projected match and its ground-truth location in the target image. Performance is reported as recall at a 10-pixel error threshold, and further analyzed according to viewpoint changes. This provides a rigorous assessment of correspondence quality across different scenes and camera poses.

*Semantic Segmentation.* A linear classifier is trained on top of the frozen patch tokens from the backbone. This classifier produces class scores at a lower spatial resolution, which are then upscaled to the original image dimensions to generate the final segmentation map. The linear layer is trained for 320,000 iterations using a single GPU. We utilize a cross-entropy loss. For optimization, we use AdamW optimizer Kingma & Ba (2014) with a learning rate of 0.001 and a batch size of two.

## A.6 BASELINE EVALUATIONS DETAILS

In this paper, we compare our pipeline with DINOv2, FiT3D, and MEF. All methods are based on a ViT architecture and provide publicly available weights for both ViT-S and ViT-B models. However, FiT3D and MEF include additional components beyond the base architecture, necessitating certain adjustments for a fair evaluation.

Unlike our approach, which utilizes features directly extracted from a finetuned DINOv2 ViT, FiT3D computes its metrics using a concatenation of features from their model and the original DINOv2. As a result, linear probing with FiT3D requires increasing the input dimension of the linear head by the embedding dimension to accommodate the additional channels. Notably, our results for depth estimation and semantic segmentation with FiT3D differ from those published in their paper. These discrepancies can be attributed to variations in dataset splits (except for ScanNet++, where splits are similar) and differences in training procedures—for instance, FiT3D trained probes for fewer training steps but with 8 Nvidia A100 GPUs, resulting in a larger effective batch size.

Similarly, when comparing with MEF, we had to introduce an additional convolutional layer on top of the patch tokens, as required by their pipeline. In contrast, our method does not require any such modifications and only trains the weights of the original DINOv2 model.

## A.7 DATASET DETAILS

**Training.** Our models are pretrained exclusively on the ScanNet++ Yeshwanth et al. (2023) dataset, following a similar protocol to Yue et al. (2024). Specifically, we use a training split of 280 indoor scenes for representation learning. To ensure consistency with our baseline Yue et al. (2024), we maintain the same input size at $(584 \times 876)$ for ScanNet++. Notably, this resolution is

significantly higher than ($180 \times 320$) used in MVSplat (Chen et al., 2024). The ScanNet++ dataset provides 3D annotations of the different objects in the scene, which are rendered to 2D and provide a weak supervision of semantic guidance during our training pipeline.

**Evaluation.** After finetuning, we assess the learned representations on various downstream tasks using linear probing. This involves creating a train-test split to train and evaluate the linear probes, while keeping the backbone model frozen throughout evaluation. For multi-view correspondence, no split is needed, as it is performed without additional training.

*ScanNet++:* For linear probing experiments on ScanNet++, we use the split described in Yue et al. (2024), with 230 scenes (140,451 views) for training, and report results on the validation set, which consists of 50 scenes (30,638 images).

*ScanNet:* For depth estimation and semantic segmentation on ScanNet Dai et al. (2017), we use the official `scannet_frames_25k` subset. For efficiency, we downscale the images in ScanNet by a factor of two, i.e., ($484 \times 968$). This subset is randomly split into 80% for training and 20% for validation. For multi-view correspondence, we utilize the ScanNet Pairs benchmark Sarlin et al. (2020), which comprises 1,500 curated image pairs. As this task does not involve linear probing or additional training, the entire set is used solely for evaluation.

*NYUv2:* For depth estimation and semantic segmentation, we use the train-test split provided by Suteu & neverix (2019) for training and evaluating the linear probes. For surface normal estimation, we follow the original split of 1,449 labeled images with ground-truth surface normal annotations Ladickỳ et al. (2014).

*ADE20K:* For semantic segmentation evaluation of the ADE20K dataset Zhou et al. (2017), we follow the official train-validation split.

*PASCAL VOC:* We further benchmark semantic segmentation on the PASCAL VOC dataset Everingham et al. (2015), adhering to the official train-test split.

*KITTI:* For monocular depth estimation, we utilize the KITTI dataset Geiger et al. (2013). Due to the absence of an official split for the depth selection subset, we generate one by randomly assigning 85% of the images to train and 15% to validation using a custom script, and use these splits for both training and evaluating the linear probes.

## A.8 LIMITATIONS

Our approach, while effective within its design scope, has two main limitations. First, the quality of our supervisory signal depends directly on the performance of the feed-forward 3DGS model: when this reconstruction is suboptimal or inaccurate, the supervision is degraded and overall performance can suffer. Second, our training currently relies on multiview image datasets, which are relatively limited in availability and diversity. Enabling training on video data would allow us to leverage substantially larger and more diverse data sources.

## A.9 USE OF LARGE LANGUAGE MODELS (LLMS)

Large Language Models (LLMs) were utilized during the development of this work to assist with the identification of related research and to provide support in the writing process, including drafting and refining sections of the manuscript. All substantive scientific contributions, experimental design, and analysis were performed by the authors.

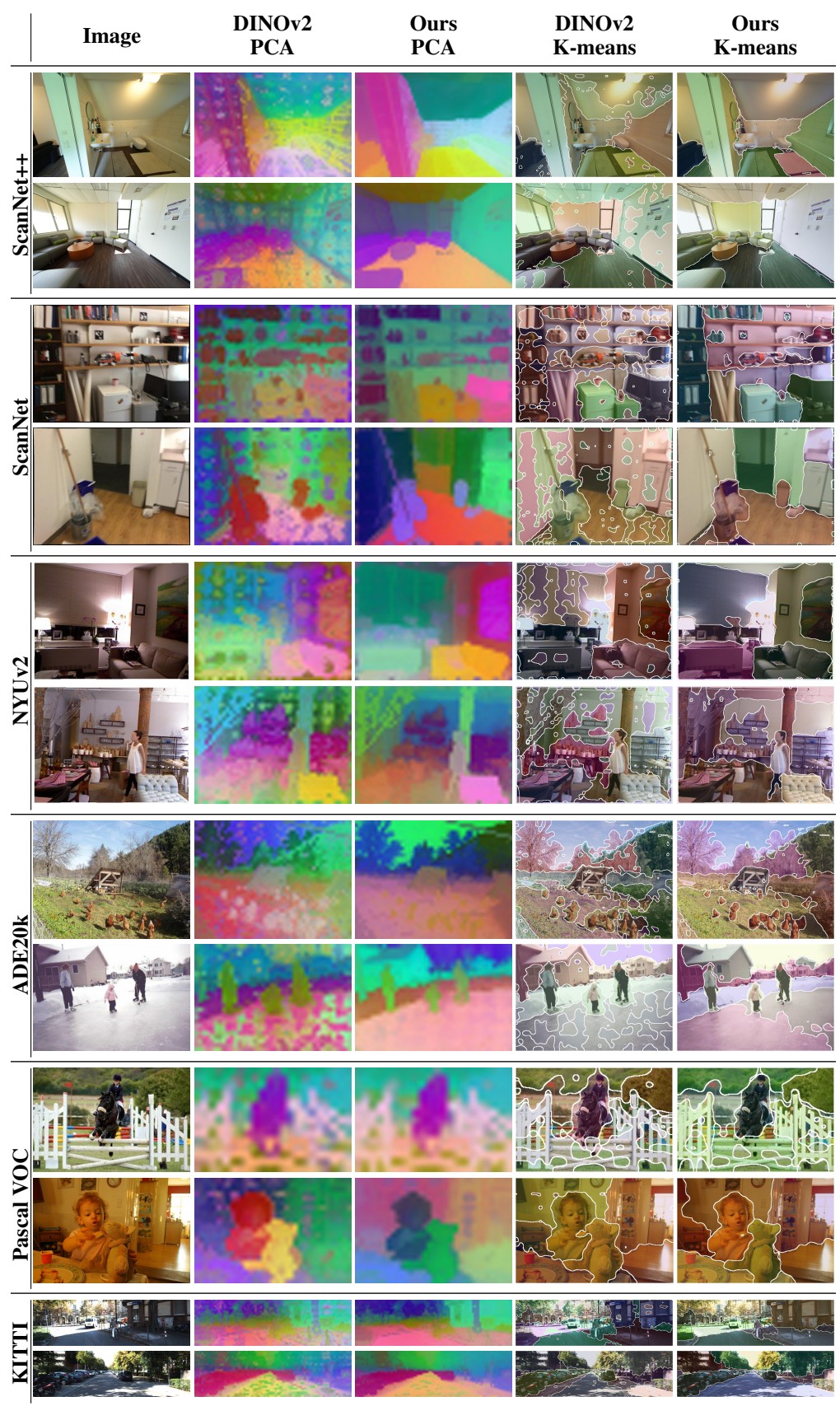

Figure 14: Qualitative visualization of DINOv2 features vs. ours on different datasets using a ViT-Small backbone. We visualize single-image features using PCA and K-means.

