# OpenReview forum: "Splat and Distill: Augmenting Teachers with Feed-Forward 3D Reconstruction For 3D-Aware Distillation"
_ICLR.cc/2026/Conference — ICLR 2026 Poster_

### Official Review · Reviewer_sbNt · 2025-10-26

**Soundness:** 3
**Presentation:** 2
**Contribution:** 3
**Rating:** 6
**Confidence:** 3

**Summary:**

The work proposes a distillation framework to enhance 3D awareness of a self-supervised encoder. The idea is to leverage a feed-forward model for multi-view 3DGS inference (pre-trained) to synthesise a 3D-consistent target for a student network. Specifically, an EMA teacher uses segmentation masks to fuse its patch-level representation into 3DGS by guided usampling. The resulting high-resolution feature grid is downsampled for student distillation.
Experiments on six datasets demonstrate notable improvement of the base model across four tasks: depth estimation, multi-view correspondence, surface normal estimation and semantic segmentation.

**Strengths:**

* Creating a 3DGS representation on-the-fly for 3D-aware feature distillation is a very nice idea.
* The experimental scope and the demonstrated results are laudable. The approach demonstrates consistent and often significant improvement across all tasks.
* The approach is relatively simple and leads to representations, which can be used standalone (without the need for concatenating them with the baseline features in evaluation).

**Weaknesses:**

* Overall, it is unclear where the gains come from. Conceptually, the approach designs a view-invariant representation and uses segmentation masks. This explains the gains on semantic segmentation, which requires view-invariance, but not the 3D-aware tasks (e.g. depth), where the representation is covariant with the camera pose.
* The ablation study is not very informative, also in the context of the above point. Even the worst configuation here (“without blending”) already outperforms the baseline DINOv2 model by a significant margin. All other configuations lead only to a marginal decrease in the task accuracy. I would like to see the most basic configuration instead: a fixed teacher model, without masked upsampling and blending.
* Related to the above points, introduction stresses “a least-squares” compromise from previous work, citing multi-view inconsistency of the base feature representation. The argument seems to  imply that there is no feature averaging in this work. However, this is exactly what Eqs. (2,4) do. The only difference is that this averaging occurs within each segment, which is not particularly critical (as experiment in Tab. 5B shows)

**Questions:**

* What aspect of the framework design encourages improvement on view co-variant tasks (e.g. depth, surface normal estimation)?
* How would the most basic variant of the proposed framework (no mask-guided usampling, no blending, no EMA) perform?
* How does the approach work around the issues of a “least-squares compromise” and not “enforcing feature similarity at corresponding points” from previous work?

---

> ### Author Response · Authors · 2025-11-20
> **Response (Part 1)**
>
> We thank the reviewer for their thoughtful and overly positive evaluation. We appreciate the recognition of our on-the-fly 3DGS distillation as a "very nice idea" and the acknowledgment of our consistent performance gains and laudable experimental scope. We address these points below:
>
> **"Conceptually... the approach designs a view-invariant representation... not suitable for covariant tasks (e.g. depth)."**
> We thank the reviewer for raising this critical point. We clarify that our method enforces **3D consistency**, which is fundamentally distinct from strict **view invariance**. Our learned features do not collapse into a "flat," view-invariant representation; rather, they encode a high-dimensional signal containing both invariant (semantic) and covariant (geometric) information.
>
> Our results suggest these signals are linearly decodable to a significantly higher degree than in baselines. A linear probe on our features extracts precise depth (ScanNet++ RMSE 0.3299, Table 1) more accurately than on DINOv2 (RMSE 0.3777) and other baselines, while a separate probe successfully extracts semantics more accurately than baselines. This indicates that our method enhances the linear accessibility of the covariant geometric signal without "overwriting" the invariant semantic signal.
>
> **Why does the framework encourage this?** The gain in covariant tasks (depth, normals) stems directly from the **Feed-Forward 3D Reconstruction module**. The rasterization of 3D Gaussians is inherently view-dependent. A feature at a specific pixel is an aggregate of multiple 3D Gaussians intersecting that ray. As the viewing angle changes, the set of intersecting Gaussians, their sorting order, and their projection weights change. Consequently, corresponding pixels in two views aggregate the same underlying 3D information but with different **projection weights**, naturally creating a signature that covaries with camera pose rather than remaining static.
>
> **Empirical Verification:** To confirm that our features do not collapse into a single invariant vector, we analyzed feature distances at corresponding ground-truth 3D locations across views:
> * **Non-Zero Variance:** We measured the average L2 distance between features at corresponding pixels in context views vs. rendered target views [over 14 images from ScanNet++]. For a "Basic" variant (without mask-aware upscaling and feature blending), the average distance is **16.62**. For our "Full" approach, the distance is **23.65**. A strictly view-invariant representation would yield a distance near zero.
> * **Structured Covariance:** This variance is not noise. The fact that linear probes achieve improved performance on **Surface Normals** over baselines (for ViT-Small on NYU, RMSE of **28.93** compared to **30.99** of vanilla DINOv2, derived from Table 2) shows that this variation is covariant with camera pose in a linearly accessible way.
> * **Simultaneous Correspondence:** Despite this necessary variance, the features remain semantically close enough for zero-shot correspondence (Figure 7), where our method outperforms baselines.
>
> **"I would like to see the most basic configuration instead: a fixed teacher model, without masked upsampling and blending."**
> We thank the reviewer for this suggestion. We ran the requested ablation (Fixed Teacher + No Mask-Aware Upsampling + No Blending) on ScanNet++ (ViT-Small). We have added this comparison into our manuscript.
>
> **Results:**
>
> | Method | Depth (abs-rel / RMSE) $\downarrow$ | Seg. (aAcc / mIoU / mAcc) $\uparrow$ |
> | :--- | :---: | :---: |
> | Vanilla DINOv2 | 0.2811 / 0.3777 | 80.23 / 29.54 / 39.11 |
> | Ours (Basic Variant) | 0.2741 / 0.3520 | 81.80 / 30.66 / 40.47 |
> | **Ours (Full SnD)** | **0.2421** / **0.3299** | **83.84** / **31.78** / **41.42** |
>
> **Analysis:**
> * **3D Supervision alone yields gains:** Even in this most basic form, our method outperforms the DINOv2 baseline (Depth RMSE 0.3520 vs. 0.3777). This confirms that distilling feed-forward 3D reconstruction provides a meaningful, standalone 3D-aware signal.
> * **Components are Non-Marginal:** The "Basic Variant" (0.3520) lags significantly behind our "Full Model" (0.3299). This gap validates that our architectural choices are integral, not marginal. Specifically, the iterative teacher update (EMA) is critical: as shown in Table 5 (Ablation D), simply freezing the teacher—even with mask refinements—results in an RMSE of 0.3444. Enabling the iterative update reduces this further to 0.3299.

---

> ### Author Response · Authors · 2025-11-20
> **Response (Part 2)**
>
> **"Introduction stresses 'least-squares' compromise... implies no feature averaging... However, this is exactly what Eqs. (2,4) do."**
> We thank the reviewer for this important observation. Indeed, the reviewer is correct that our method involves feature averaging, particularly via the spatial smoothing in Eqs. (2, 4). However, the "least-squares compromise" noted in our introduction refers to the averaging of inconsistent features across multiple views at corresponding locations, rather than the spatial averaging within a single frame (Eqs. 2, 4). This will be fully clarified.
>
> To mitigate this specific **multi-view** averaging effect, we employ two main strategies:
> 1.  **Soft vs. Hard Loss:** Unlike methods that regress features using MSE (which mathematically encourages averaging), we use a "soft" cross-entropy objective over learnable prototypes. This allows the student to match the underlying semantic distribution without overfitting to the exact mean of the teacher's artifacts.
> 2.  **Iterative Evolution:** Our teacher is not a fixed anchor (as in Fit3D). Through EMA updates, the teacher evolves. While averaging can still occur, this dynamic process prevents the model from getting "locked" into the initial inconsistencies of the pretrained features.
>
> We further note that both the iterative refinement and soft distillation play a similar role as in SSL approaches (e.g., DINOv2), where the goal is to match feature distributions rather than exact values, preventing the model from locking onto 3D inconsistencies.
>
> While we acknowledge that this strategy may not completely eliminate the compromise, our empirical evidence suggests it significantly mitigates it. As shown in Table 5(D) (Ablation of Frozen vs. Learnable Teacher) and the "Basic Variant" ablation provided above, both the iterative update and the soft distillation strategies result in improved performance over fixed baselines.
>
> To summarize, we appreciate the thoughtful and overall positive feedback from reviewer sbNt. We believe our answers address the points raised and further validate the contributions of our work and have updated our manuscript accordingly. We will be happy to address any further concerns the reviewer may have.

---

### Official Review · Reviewer_MPDZ · 2025-10-29

**Soundness:** 3
**Presentation:** 3
**Contribution:** 3
**Rating:** 2
**Confidence:** 4

**Summary:**

The method extends DINO-style self-distillation by integrating a feed-forward 3D reconstruction pipeline. Two context views are used to build a 3D Gaussian scene via a pretrained model (MVSplat), while the teacher’s 2D features are lifted onto this geometry to form 3D feature Gaussians. These are rendered from a novel viewpoint to generate a 3D-aware supervisory map for the student. Mask-aware feature lifting preserves semantic boundaries, and semantic blending regularizes noisy regions. The student, given only the target image, learns to match the rendered teacher features through a cross-entropy loss, with the teacher updated via EMA. This yields 2D features enriched with 3D geometric awareness.

**Strengths:**

- The method demonstrates clear performance gains over DINOv2, the baseline framework it builds upon.
- The use of a feed-forward 3D reconstruction module to inject geometric awareness into 2D foundation models seems a promising and timely research direction.

**Weaknesses:**

The evaluation setup is limited, as DINOv2 is seldom used directly for downstream tasks. It typically serves as a feature backbone for task-specific heads. Hence, direct comparison against DINOv2 provides a very limited insight. Demonstrating improvements when replacing DINOv2 with the proposed method within state-of-the-art pipelines (e.g., VGGT) would make the results substantially more compelling.

Many recent methods seem to be missing from the comparisons:
- DINOv3
- Sarıyıldız, M.B., Weinzaepfel, P., Lucas, T., de Jorge, P., Larlus, D. and Kalantidis, Y., 2025. DUNE: Distilling a Universal Encoder from Heterogeneous 2D and 3D Teachers. In Proceedings of the Computer Vision and Pattern Recognition Conference (pp. 30084-30094).
- Govindarajan, H., Wozniak, M.K., Klingner, M., Maurice, C., Kiran, B.R. and Yogamani, S., 2025. CleverDistiller: Simple and Spatially Consistent Cross-modal Distillation. BMVC 2025 (on arxiv since March)
- Huang, X., Wu, J., Xie, Q. and Han, K., 2025. MLLMs Need 3D-Aware Representation Supervision for Scene Understanding. arXiv preprint arXiv:2506.01946.
- You, Y., Li, Y., Deng, C., Wang, Y. and Guibas, L., 2024. Multiview Equivariance Improves 3D Correspondence Understanding with Minimal Feature Finetuning. ICLR 2025

Minor thing:
- Section 3.3 closely resembles the feature lifting strategy used in OccamLGS; the authors should probably cite it.

**Questions:**

- How does the proposed approach perform when used as a drop-in replacement for DINOv2 within task-specific architectures such as VGGT?
- The evaluated downstream tasks (e.g., depth estimation) are interesting, but 2D foundation models are rarely applied to them directly. It would be helpful to also include results from task-specific state-of-the-art methods on the same benchmarks as reference to see where the proposed approach stands in comparison.
- How does the method compares to the missing baselines?

**Details Of Ethics Concerns:**

No concerns.

---

> ### Author Response · Authors · 2025-11-20
> **Response (Part 1)**
>
> We thank the reviewer for their thoughtful and constructive feedback. We appreciate the recognition of our method's clear performance gains over DINOv2 and the endorsement of our feed-forward 3D reconstruction approach as a "promising and timely research direction". We address points raised below:
>
> **"The evaluation setup is limited, as DINOv2 is seldom used directly..."**
> Thanks for raising this important point, which we address using three parts below:
>
> **1. Linear Probing Protocol:** We note that linear probing/zero-shot evaluation is the standard mechanism for evaluating representation learning quality (used by DINO, MAE, etc.). Crucially, for 3D-aware VFM adaptation, we adhere to the evaluation protocol established by Fit3D, which similarly considered linear probing/zero-shot evaluation. Introducing complex task-specific heads or full fine-tuning would introduce confounding factors, potentially masking the intrinsic quality of the features. By adhering to linear probing, we isolate the improvements in the representation space itself, ensuring a fair, direct comparison with the protocols established by Fit3D and DINOv2.
>
> **2. Task-Specific Backbones (VGGT):** Nevertheless, following the reviewer's suggestion, we consider the performance of the proposed approach when used as a drop-in replacement for DINOv2 within VGGT. Specifically, we evaluated SnD as a backbone within the **VGGT** pipeline for monocular depth on ScanNet++. We adapted the pre-trained VGGT to accept ViT-Small features via a learnable projection ($384\rightarrow1024$), fine-tuning only this projection and the depth head while freezing the transformer layers. We adopted this protocol because: 1. This setup scientifically isolates the quality of the backbone representations, ensuring that performance gains are attributable solely to improved 3D awareness rather than the decoder's capacity. 2. Full fine-tuning of VGGT is computationally prohibitive within the rebuttal window. We considered the same setup, using either SnD (ViT-S) or an identical Vanilla DINOv2 (ViT-S) baseline, obtaining:
>
> | Method | Depth (Abs-Rel) $\downarrow$ | Depth (RMSE) $\downarrow$ |
> | :--- | :---: | :---: |
> | VGGT with DINOv2 backbone | 0.2220 | 0.3426 |
> | **VGGT with our backbone** | **0.2117** | **0.3283** |
>
> As can be seen, using our backbone results in improved performance. We have added this comparison into our manuscript.
>
> **3. Use of Direct Features in Literature:** DINOv2 (and DINO) is used directly for downstream tasks in numerous works, without serving as a feature backbone for task-specific heads. While it is not possible to list all such works, we provide here an example subset of relevant papers.
>
>   * **Unsupervised Segmentation & Localization:** Frozen features are widely used with clustering (k-means, spectral) and label propagation for competitive unsupervised segmentation and object discovery [2, 4, 6–14].
>   * **Dense Correspondence, Pose & SLAM:** Raw patch tokens are effectively used for co-segmentation, semantic correspondence, 6D pose estimation, and loop closure via k-NN or cosine similarity [1, 3, 5, 15–19].
>
> Our method, which improves the geometric consistency of these raw features, is directly applicable to improving these works.
>
> **"Also include results from task-specific state-of-the-art methods"**
> As demonstrated in Point 2, SnD improves performance when integrated as a backbone into task-specific pipelines (VGGT). However, directly comparing our **linear probing** results to specialized SOTA methods is methodologically inconsistent. Linear probing is designed to assess the linear separability and intrinsic structure of the frozen feature space, whereas SOTA methods rely on complex, non-linear decoders trained end-to-end. Analogous to standard SSL evaluation (e.g., DINO), linear heads are used to benchmark generalizable representation quality, rather than to compete with specialized architectures optimized for a single task.
>
> **Missing Baselines**
> We thank the reviewer for bringing these relevant works to our attention. We will fully incorporate the discussion of these works below into the final manuscript.
>
> 1.  **DINOv3:** This work appeared on arXiv in August 2025 and has not been published in a peer-reviewed venue. Per ICLR policy regarding concurrent work (papers published on/after July 24, 2025), authors are not required to compare against it. However, since our SnD is a framework applicable to ViT backbones generally, we expect our geometric improvements to transfer to DINOv3 as well.

---

> ### Author Response · Authors · 2025-11-20
> **Response (Part 2)**
>
> 2.  **DUNE:** We compare DUNE on monocolor depth estimation and semantic segmentation on ScanNet++ (ViT-Small). We evaluated the official DUNE repository. They offer two model weight variants `dune_vitsmall14_336.pth`, and `dune_vitsmall14_448.pth`, and we show results on the latter, as it produced better results between the two. As shown below, SnD outperforms DUNE on 3D-aware tasks:
>
>     | Method | Depth (Abs-Rel / RMSE) $\downarrow$ | Seg. (aAcc / mIoU / mAcc) $\uparrow$ |
>     | :--- | :---: | :---: |
>     | DUNE | 0.3144 / 0.3929 | 79.64 / 25.77 / 34.74 |
>     | **Ours (SnD)** | **0.2421** / **0.3299** | **83.84** / **31.78** / **41.42** |
>
>     **Analysis:** We believe DUNE and SnD address complementary goals. DUNE excels at distilling a universal encoder from heterogeneous teachers (including 2D and 3D sources), but it also distills DINOv2 features directly, inheriting their inherent 3D inconsistencies. In contrast, SnD focuses specifically on correcting these inconsistencies via our feed-forward 3D reconstruction pipeline before distillation. We added this comparison and discussion to our manuscript.
>
>     **Future Direction:** We see an exciting opportunity to combine these approaches: using SnD to geometrically correct the teacher features *before* they are distilled by the DUNE framework. This could potentially yield an ultimate universal encoder that is broad and 3D-consistent.
>
> 3.  **CleverDistiller:** This method operates on LiDAR point clouds to learn a 3D backbone, whereas we improve 2D image encoders, which is a different task. Hence, the works are not directly comparable. Furthermore, BMVC 2025 decisions were released on July 25, 2025. Per ICLR policy, this falls into the concurrent work period, meaning we are not required to compare to it.
>
> 4.  **MLLMs Need 3D-Aware...:** This work targets text generation in MLLMs, not dense feature map extraction; the tasks are fundamentally different, and so the works are not directly comparable. Additionally, as a NeurIPS 2025 paper (decisions Sep 18, 2025), this is considered concurrent work per ICLR policy, and comparison is not required.
>
> 5.  **Multiview Equivariance... (You et al., 2024):** This corresponds to the **MEF** baseline, which we extensively compared to in our main paper, demonstrating that SnD achieves significantly higher performance.
>
> **OccamLGS:** Thanks, we have now added this citation to our manuscript in Sec. 3.3.
>
> To summarize, we appreciate the thoughtful and constructive feedback from reviewer MPDZ. We believe our answers address the points raised and further validate the contributions of our work and have updated our manuscript accordingly. We will be happy to address any further concerns the reviewer may have.
>
> **References:**
> [1] Amir et al., "Deep ViT Features as Dense Visual Descriptors", arXiv 2021.
> [2] Docherty et al., "Upsampling DINOv2 Features for Unsupervised Vision Tasks and Weakly Supervised Materials Segmentation", arXiv 2024.
> [3] Örnek et al., "FoundPose: Unseen Object Pose Estimation with Foundation Features", ECCV 2023.
> [4] Wang et al., "Self-Supervised Transformers for Unsupervised Object Discovery Using Normalized Cut", CVPR 2022.
> [5] Vanyan et al., "Analyzing Local Representations of Self-supervised Vision Transformers", arXiv 2023.
> [6] Cheung et al., "A Lightweight Clustering Framework for Unsupervised Semantic Segmentation", arXiv 2023.
> [7] Melas-Kyriazi et al., "Deep Spectral Methods: A Surprisingly Strong Baseline for Unsupervised Semantic Segmentation and Localization", CVPR 2022.
> [8] Ypsilantis et al., "Co-Segmentation without any Pixel-level Supervision with Application to Large-Scale Sketch Classification", ACCV 2024.
> [9] Rewatbowornwong et al., "Zero-guidance Segmentation Using Zero Segment Labels", ICCV 2023.
> [10] Barsellotti et al., "Talking to DINO: Bridging Self-Supervised Vision Backbones with Language for Open-Vocabulary Segmentation", arXiv 2024.
> [11] Siméoni et al., "Localizing Objects with Self-Supervised Transformers and No Labels", BMVC 2021.
> [12] Caron et al., "Emerging Properties in Self-Supervised Vision Transformers", ICCV 2021.
> [13] Stojnić et al., "LPOSS: Label Propagation Over Patches and Pixels for Open-vocabulary Semantic Segmentation", CVPR 2025.
> [14] Wang et al., "TokenCut: Segmenting Objects in Images and Videos With Self-Supervised Transformer and Normalized Cut", TPAMI 2022.
> [15] Simoncini et al., "No Train, all Gain: Self-Supervised Gradients Improve Deep Frozen Representations", arXiv 2024.
> [16] Gonzalez et al., "Multi-modal Loop Closure Detection with Foundation Models in Severely Unstructured Environments", 2025.
> [17] Wagner et al., "Oh-A-DINO: Understanding and Enhancing Attribute-Level Information in Self-Supervised Object-Centric Representations", 2025.
> [18] Huang et al., "DIVE: Taming DINO for Subject-Driven Video Editing", arXiv 2024.
> [19] Shtedritski et al., "Learning Universal Semantic Correspondences with No Supervision and Automatic Data Curation", ICCVW 2023.

---

> > ### Comment · Reviewer_MPDZ · 2025-11-24
> >
> > Thanks for the detailed answer and new experiments. I am happy with the answer and willing to increase my rating.

---

> > > ### Author Response · Authors · 2025-11-25
> > > **Thank you**
> > >
> > > We thank the reviewer for the support and for updating the score.

---

### Official Review · Reviewer_C1do · 2025-11-01

**Soundness:** 3
**Presentation:** 3
**Contribution:** 2
**Rating:** 6
**Confidence:** 3

**Summary:**

This paper introduces SnD, a new pipeline to improve the 3D awareness of 2D foundation features. Similar to previous methods such as FiT3D, SnD uses 3DGS as 3D representation to improve 3D awareness. Instead of per-scene optimization, SnD uses feed-forward 3DGS (MVSplat) to improve efficiency. In addition, a student-teacher framework, similar to DINO, is used to distill the 3D awareness from teacher model. SnD outperforms baselines in various downstream tasks.

**Strengths:**

1. The paper is well-written and easy to follow.

2. Based on MVSplat, a feed-forward 3DGS method, SnD is more efficient than previous methods that require optimization.

3. SnD outperforms baselines in various downstream tasks, including depth estimation, normal estimation, and semantic segmentation.

**Weaknesses:**

1. The authors should add an ablation study without student-teacher framework. For example, a potential experiment is finetuning a model with feature rendering loss, similar to Fit3D. Currently, it is unclear to me why student-teacher framework is necessary.

2. The improvement that mask-aware feature lifting brings is not explicit.

3. Quantitative evaluation of multi-view feature correspondence should be added, instead of using visualization only.

**Questions:**

1. In Table 1, I suggest clarifying that SnD is finetuned on ScanNet++.

2. Semantic mask: As shown in Fig. 11, the mask used is actually instance mask, instead of semantic mask (two monitors have different masks). This should be clarified in Sec. 3.3.

3. Typos:
    * L123: of are -> are
    * L294: teacher’s are -> teacher’s parameters are

---

> ### Author Response · Authors · 2025-11-20
> **Response (Part 1)**
>
> We thank the reviewer for their constructive evaluation. We appreciate the recognition of our paper’s clarity, the superior efficiency of our feed-forward framework compared to optimization-based methods, and our consistent performance gains across downstream tasks. The feedback regarding ablations and metrics has notably strengthened our work. We address the specific comments below:
>
>
> **1. Student-teacher ablation study.**
> We thank the reviewer for noting this important ablation. We follow the reviewer's exact suggestion: we finetune a model with a feature rendering loss, as in Fit3D, rather than the student-teacher distillation framework. To allow for a fair comparison, we keep all other components the same (including mask-aware upsampling and feed-forward reconstruction).
>
> The table below summarizes the results on a ViT-Small variant on ScanNet++, evaluating the **Feature Rendering** baseline (requested by the reviewer), the **Frozen Teacher** ablation (Ablation D from our paper), and our **Full SnD** model:
>
> | Method | Teacher Update | Loss Type | Depth (Abs-Rel $\downarrow$ \| RMSE $\downarrow$) | Seg. (aAcc $\uparrow$ \| mIoU $\uparrow$ \| mAcc $\uparrow$) |
> | :--- | :--- | :--- | :--- | :--- |
> | Vanilla DINOv2 | N/A | N/A | 0.2811 \| 0.3777 | 80.23 \| 29.54 \| 39.11 |
> | Ours (Basic Variant) | Fixed | Feature Rendering | 0.2484 \| 0.3430 | 83.06 \| 31.40 \| 40.97 |
> | Frozen Teacher (Ablation D) | Fixed | Distillation | 0.2500 \| 0.3444 | 83.34 \| 31.90 \| 41.88 |
> | **Ours (Full SnD)** | **EMA** | **Distillation** | **0.2421 \| 0.3299** | **83.84 \| 31.78 \| 41.42** |
>
> As shown, while the basic Feature Rendering variant improves over Vanilla DINOv2, our Full SnD framework yields the best overall performance. We have updated our manuscript with the above.
>
> **Why is teacher-student distillation important?**
> In Fit3D, the teacher is frozen, treating the initial features as a fixed target. In contrast, our framework employs an iterative update of the teacher coupled with a specialized distillation objective. This is crucial for two reasons:
>
> * **Iterative Refinement (EMA):** Using an exponential moving average (EMA) creates a beneficial contrast between the student's rapid updates (via gradient descent) and the teacher's stable evolution. As the student improves its 3D consistency, the teacher slowly adopts these improvements, creating a virtuous cycle where the feed-forward 3D reconstruction module continuously refines the teacher’s output, which then improves the student. This iterative process then continues. This is empirically validated by comparing **Frozen Teacher** with **Full SnD**: enabling the EMA update drives the significant gain in Depth estimation (RMSE reduces from $0.3444$ to $0.3299$), confirming that co-evolving the teacher is vital for instilling robust geometric awareness.
> * **Artifacts Mitigation (Soft Distillation Objective):**
>     * **Semantics (Segmentation):** The feature rendering loss of Fit3D forces the student to strictly reproduce the teacher’s raw splatted output. We observe that this causes the model to overfit to rendering artifacts, resulting in lower segmentation performance compared to distillation  (mIoU $31.90$ vs $31.40$). Our 'soft' distillation objective, which matches distributions over learnable prototypes, acts as a semantic filter, effectively ignoring this geometric noise.
>     * **Geometry (Depth):** With a fixed teacher, feature rendering is marginally better for Depth than Distillation ($0.3430$ vs $0.3444$).
>     * **Synthesis:** By combining Distillation (to filter artifacts) with EMA (to improve the target), our Full SnD model overcomes the limitations of both baselines, achieving the state-of-the-art result in Depth ($0.3299$) while maintaining high semantic performance.
>
> **2. “The improvement that mask-aware feature lifting brings is not explicit”**
> While the quantitative gains are modest (Depth RMSE improves $0.3309 \to 0.3299$, Segmentation mIoU $31.46 \to 31.78$ in Table 5), this component is primarily included to ensure qualitative fidelity. Unlike standard bilinear upscaling, which causes feature 'bleeding' across boundaries, mask-aware lifting enforces sharper semantic transitions at object edges. This effect is visualized in **Appendix Figure 11**, which shows that mask-aware upscaling produces cleaner boundaries compared to bilinear interpolation.
>
> **3. "Quantitative evaluation of multi-view feature correspondence should be added..."**
> We respectfully point out that this quantitative evaluation is already provided in **Figure 7** of the main paper. This figure reports the Recall@10px metric across varying viewpoint changes (ranging from $0^\circ$ to $180^\circ$) for both ViT-Small and ViT-Base backbones. As shown, our method consistently improves recall over all baselines (DINOv2, MEF, Fit3D) across all angular ranges.

---

> ### Author Response · Authors · 2025-11-20
> **Response (Part 2)**
>
> **Clarifications and Typos:**
> We thank the reviewer for their detailed reading. We updated the manuscript to explicitly state that SnD is fine-tuned on ScanNet++ (Table 1), and clarify in Sec. 3.3 that we utilize instance masks (as correctly noted, this ensures separation of distinct objects like the monitors in Fig. 11), and fix the typos at L123 and L294.
>
> To summarize, we appreciate the thoughtful and overall positive feedback from reviewer c1do. We believe our answers address the points raised and further validate the contributions of our work, and have updated our manuscript accordingly. We will be happy to address any further concerns the reviewer may have.

---

### Author Response · Authors · 2025-11-30
**Summary of Reviewer Discussion and Consensus**

We thank the reviewers for their detailed and constructive feedback. This engagement allowed us to strengthen our work significantly. We also thank the AC for their time in assessing our work. Below is a summary of the discussion:

### 1. Status of Consensus (Pre-score reversal)
First, we highlight that before the score reversion, we reached a **unanimous positive consensus** with all reviewers.

* **Initial Status:** The paper received positive ratings (6, 6) from **Reviewers C1do** and **sbNt**, who commended the method's novelty and performance.
* **Resolution with Reviewer MPDZ:** Following extensive experimental updates, **Reviewer MPDZ** (initial score of 2) explicitly confirmed the resolution of their concerns.
* **Final Evaluation (Nov 24):** Reviewer MPDZ stated:
    “Thanks for the detailed answer and new experiments. I am happy with the answer and willing to increase my rating”.
    This was followed by an explicit increase in the score (**score of 6**) on Nov 24th.

### 2. Consensus on Strengths
Reviewers were consistent in recognizing the paper's methodological novelty and efficiency:

* **Novelty:** The feed-forward 3D reconstruction approach was highlighted as a "promising and timely research direction" (**Reviewer MPDZ**), and a "very nice idea" (**Reviewer sbNt**).
* **Performance:** Reviewers confirmed that the method is "more efficient than previous methods" and "outperforms baselines in various downstream tasks" (**Reviewer C1do**). They further noted that it demonstrates "consistent and often significant improvement across tasks" (**Reviewer sbNt**) and shows "clear performance gains" (**Reviewer MDPZ**).
* **Clarity:** The manuscript was praised as "well-written and easy to follow" (**Reviewer C1do**) with a "laudable" experimental scope (**Reviewer sbNt**).

### 3. Detailed Resolution of Concerns
In our revision, we addressed all reviewer concerns through extensive new experiments and clarifications, now incorporated into the manuscript:

### **Reviewer C1do**
* **Ablation of Student-Teacher:** We implemented the requested "Feature Rendering" ablation (finetuning without distillation). Results confirm that while this variant is superior to vanilla DINOv2, our full framework outperforms it, validating the necessity of the student-teacher framework.
* **Mask-Aware Lifting:** We clarified that mask-aware lifting is essential for qualitative fidelity, enforcing sharper semantic transitions at object edges (visualized in Appendix Figure 11).
* **Quantitative Correspondence:** We pointed to Figure 7, which provides the requested quantitative evaluation, showing SnD (our method) outperforms baselines.
* **Clarifications & Typos:** We fully addressed all requests for clarifications and typos.

### **Reviewer MPDZ**
* **Evaluation Setup & Downstream Tasks:** We addressed the concern that "DINOv2 is seldom used directly" in three ways:
    * *Standard Protocol:* We clarified that linear probing/zero-shot evaluation is the established standard (e.g., Fit3D, DINO) for assessing representation quality.
    * *Task-Specific Backbone:* We integrated SnD as a backbone in VGGT. Results show SnD improves over DINOv2 backbone for depth estimation, validating performance in complex architectures.
    * *Literature Evidence:* We provided a list of prior works (e.g., unsupervised segmentation, dense correspondence) where DINOv2 features are used directly.
* **Resolution of Missing Baselines:** We provided a detailed analysis of all suggested works:
    * *DUNE (Implemented):* We compared against DUNE on ScanNet++. SnD significantly outperforms DUNE on 3D-aware tasks. We discussed how DUNE distills DINOv2 inconsistencies, whereas SnD corrects them.
    * *MEF (Clarified):* We clarified that "Multiview Equivariance (You et al., 2024)" is the MEF baseline already compared in our main paper, where SnD achieves significantly higher performance.
* **Concurrent/Out-of-Scope Works:**
    * *DINOv3:* Identified as concurrent (Aug 2025). We noted that SnD is a framework likely transferable to DINOv3.
    * *CleverDistiller:* Identified as concurrent (July 2025) and domain-distinct (LiDAR point clouds vs. 2D images).
    * *MLLMs Need 3D-Aware:* Identified as concurrent (NeurIPS 2025) and task-distinct (text generation vs. dense features).
    * *OccamLGS:* Added the requested citation in Section 3.3.

### **Reviewer sbNt**
* **"Basic Variant" Ablation:** We evaluated the requested basic configuration (fixed teacher, no mask-aware upsampling, no blending). While this variant improves over vanilla DINOv2, our full model outperforms it.
* **Viewpoint Covariance:** We provided empirical analysis confirming our features are covariant (not view-invariant), preserving the necessary 3D consistency for geometry tasks like depth/normal estimation. We also clarified why this is the case.
* **Feature Averaging:** We clarified how our distillation loss and iterative teacher evolution mitigate feature averaging artifacts.

---

### Meta-Review · Area_Chair_XVW4 · 2026-01-06

**Summary:**

The paper introduces a framework to infuse 3D awareness into 2D vision foundation models like DINOv2. The core idea is to augment a teacher network with a fast, feed-forward 3D reconstruction pipeline (based on MVSplat). Teacher features are lifted into explicit 3D Gaussian representations and splatted onto novel viewpoints to supervise a student model. This process distills geometrically grounded knowledge, enhancing performance on downstream tasks such as monocular depth estimation, surface normal estimation, and semantic segmentation.

One reviewer initially questioned the evaluation setup, but the use of linear probing is indeed a standard protocol for assessing representation quality. There were also concerns regarding missing baselines (e.g. DUNE), but it has been resolved by the new experiments added during rebuttal. And some other works were identified as concurrent or out of scope. Moreover, the authors also added more ablation studies to isolate the gains from different components.

I think the authors provided a comprehensive rebuttal that addressed key concerns raised by reviewers, leading to a consensus in the end. The reviewer who initially assigned a low score explicitly committed to raising their rating (indeed raised from 2 to 6 during the discussion period).

**Reviewer Concerns:**

All concerns have been addressed.

**Reviewer Scores:**

Two reviewers who gave 6 will probably keep their rating, while the one who gave 2 has increased to 6.

---

### Decision · Program_Chairs · 2026-01-26

Accept (Poster)